# OVTR: End-to-End Open-Vocabulary Multiple Object Tracking with Transformer

**Jinyang Li**[†]    **En Yu**[†]    **Sijia Chen**    **Wenbing Tao**[*]
Huazhong University of Science and Technology
{jinyangli, yuen, sijiachen, wenbingtao}@hust.edu.cn

## Abstract

Open-vocabulary multiple object tracking aims to generalize trackers to unseen categories during training, enabling their application across a variety of real-world scenarios. However, the existing open-vocabulary tracker is constrained by its framework structure, isolated frame-level perception, and insufficient modal interactions, which hinder its performance in open-vocabulary classification and tracking. In this paper, we propose **OVTR** (End-to-End **O**pen-**V**ocabulary Multiple Object **T**racking with T**R**ansformer), the first end-to-end open-vocabulary tracker that models motion, appearance, and category simultaneously. To achieve stable classification and continuous tracking, we design the **CIP** (**C**ategory **I**nformation **P**ropagation) strategy, which establishes multiple high-level category information priors for subsequent frames. Additionally, we introduce a dual-branch structure for generalization capability and deep multimodal interaction, and incorporate protective strategies in the decoder to enhance performance. Experimental results show that our method surpasses previous trackers on the open-vocabulary MOT benchmark while also achieving faster inference speeds and significantly reducing preprocessing requirements. Moreover, the experiment transferring the model to another dataset demonstrates its strong adaptability. Models and code are released at https://github.com/jinyanglii/OVTR.

## 1 Introduction

Critical to video perception, multiple object tracking (MOT) can currently be applied to various downstream tasks such as autonomous driving and video analysis (Bashar et al., 2022). Dominant MOT methods are primarily trained to track closed-vocabulary categories, limiting their ability to generalize to unseen categories and a broader range of scenarios. Clearly, such approaches do not offer the ultimate solution for human-like video perception intelligence, as humans can perceive and track unseen dynamic objects in an open-world context. To address this, the open-vocabulary multiple object tracking (OVMOT) task (Li et al., 2023) was proposed, where models are expected to identify and track novel categories in a zero-shot manner, aligning better with real-world demands, such as more comprehensive video understanding, smart cities, and autonomous driving.

Recently, as numerous open-vocabulary detection (OVD) (Gu et al., 2021; Du et al., 2022; Lin et al., 2022; Wu et al., 2023b; Zang et al., 2022) methods have emerged, researchers (Li et al., 2023) have extended OVD into the tracking domain by integrating open-vocabulary detectors with appearance-based associations, as illustrated in Fig. 1. This approach utilizes OVD and data augmentation to improve appearance-based association learning. However, it encounters three problems: (1) Classification and tracking are merely cobbled together, rather than collaborating effectively. Treating classification independently in each frame causes instability in category perception and hinders the reuse of previous predictions in subsequent frames. (2) From the framework perspective, appearance-based association struggles to adapt to the diverse environments typical of OVMOT task. Moreover, the tracking-by-OVD framework inevitably relies on complex post-processing and anchor generation, which reduce inference speed and necessitate hand-designed operations based on scene-specific prior knowledge, making it difficult to adapt in an open-world context. (3) Achieving

---

[†]Equal Contribution.
[*]Corresponding author.

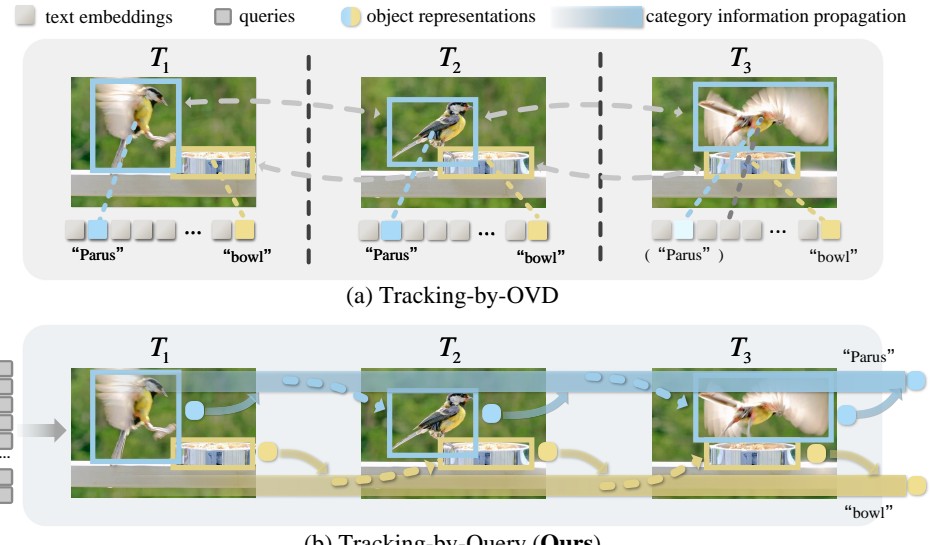

Figure 1: **Comparison of tracking-by-OVD and our method.** Tracking-by-OVD predicts each frame independently, making classification and association susceptible to changes in appearance. In contrast, our method, OVTR, propagates location, appearance, and category information from the current frame to subsequent frames, creating a stable, continuously updated information flow. This flow serves as a prior, aiding in capturing the corresponding target in future frames.

open-vocabulary capabilities requires significant time to use a pre-trained image encoder to extract numerous object embeddings that implicitly contain unseen categories from datasets, which prolongs the model development cycle. Despite this, the performance of open-vocabulary classification remains insufficient.

Regarding problem (1), an iterative tracking mechanism (Zeng et al., 2022) can be leveraged to propagate category information to subsequent frames. For problem (2), existing closed-set, end-to-end transformer-based tracking methods (Zeng et al., 2022; Zhang et al., 2023; Gao & Wang, 2023; Yu et al., 2023b) focus on achieving sustained tracking in complex scenarios. With their elegant framework, these methods eliminate the need for complex post-processing and explicit tracking associations, showing potential for cross-temporal modeling of targets in specific scenarios. Adopting such a structure effectively mitigates these challenges. Taken together, an iterative, transformer-based, end-to-end tracker such as MOTR (Zeng et al., 2022) can serve as a suitable foundation.

Based on the iterative tracking mechanism, we propagate category information across multiple frames, which we refer to as *the category information propagation (CIP) strategy*. By converting predicted category information into priors for subsequent tracking, it establishes a higher-level flow that enables the continuous tracking of corresponding targets. Additionally, due to the introduction of attention mechanism, potential interference among differing information must also be considered. To address this, we designed two *attention protection strategies*. These strategies construct masks based on category prediction distributions and the arrangement of queries, effectively intervening to ensure harmonious cooperation between classification and tracking.

To address problem (3), we design our method from the perspective of knowledge distillation and deep modality interaction. Considering that knowledge distillation between two models processing the same modality may be more effective, we use the CLIP (Radford et al., 2021) image encoder, a source of open-vocabulary, to guide our model in acquiring open-vocabulary generalization capabilities in image perception, aligning intermediate outputs (termed *aligned queries*) with CLIP image embeddings. Meanwhile, given the shortcomings in prior work regarding multimodal fusion, we design attention-based fusion in both the encoder and decoder. Particularly, in the decoder, we propose a dual-branch structure. One branch serves as the medium through which the CLIP image encoder guides our model, while the other branch allows the aligned queries to interact with text embeddings from the CLIP text encoder to focus on category information, yielding features rich in category information for open-vocabulary classification. Thus, through *the dual-branch decoder* design, we avoid cumbersome preprocessing while achieving improved open-vocabulary performance.

Incorporating all the aforementioned designs, we propose **OVTR** (End-to-End **O**pen-**V**ocabulary Multiple Object **T**racking with T**R**ansformer), the first end-to-end open-vocabulary tracker that integrates motion, appearance, and category modeling. Experimental results on the TAO (Dave et al., 2020) dataset demonstrate that OVTR outperforms state-of-the-art methods on the TETA (Li et al., 2022b) metric. Notably, on the validation set, OVTR exceeds OVTrack by 12.9% on the novel TETA, while on the test set, it surpasses OVTrack by 12.4%. Additionally, in the KITTI (Geiger et al., 2012) transfer experiment, OVTR outperforms OVTrack by 2.9% on the MOTA metric.

To summarize, our main contributions are listed as below:

- We propose the first end-to-end open-vocabulary multi-object tracking algorithm, introducing a novel perspective to the OVMOT field, achieving faster inference speeds, and possessing strong scalability with potential for further improvement.
- We propose the category information propagation (CIP) strategy to enhance the stability of tracking and classification, along with the attention isolation strategies that ensure open-vocabulary perception and tracking operate in harmony.
- We propose a dual-branch decoder guided by an alignment mechanism, empowering the model with strong open-vocabulary perception and multimodal interaction capabilities while eliminating the need for time-consuming preprocessing.

## 2 RELATED WORK

**Open-Vocabulary MOT.** Multi-object tracking (MOT) is the task of identifying and following multiple objects across consecutive frames in a video. Many trackers exhibit powerful capabilities and performance (Wojke et al., 2017; Zhang et al., 2022b; Yu et al., 2022; Chen et al., 2024a;b), and MOT has been scaled up to handle a broader range of categories, enabling it to operate in more diverse environments. Dave et al. (2020) introduced the TAO benchmark, which includes more than 800 categories and emphasizes MOT performance on the long-tail distribution of a large number of categories. TETA (Li et al., 2022b) decomposes tracking evaluation into three sub-factors: localization, association, and classification, allowing for comprehensive benchmarking of tracking performance even under inaccurate classification, with TETer (Li et al., 2022b) achieving strong results on this evaluation. OVTrack (Li et al., 2023) leverages the open-vocabulary detector ViLD (Gu et al., 2021), combining appearance-based association for open-vocabulary tracking, and uses diffusion models to generate LVIS (Gupta et al., 2019) image pairs for training associations. MASA (Li et al., 2024) learns a general appearance matching model using a large number of unlabeled images. SLAck (Li et al., 2025) integrates semantic and location information to enhance tracking performance.

**Transformer-based MOT.** Association-based trackers have not yet fully realized end-to-end tracking, as they still depend on post-processing and explicit matching strategies. In contrast, the Transformer model holds advantages as a framework for implementing end-to-end MOT. TransTrack (Sun et al., 2020) adopts a decoupled network and IoU matching to achieve query-based detection and tracking. TrackFormer (Meinhardt et al., 2022) and MOTR (Zeng et al., 2022) reformulate tracking as a sequence prediction task, where each trajectory is represented by a track query. MOTRv2 (Zhang et al., 2023) incorporates an additional object detector to generate proposals serving as anchors, providing detections for MOTR. MOTRv3 (Yu et al., 2023b) refines the label assignment process by employing a release-fetch supervision method. Additionally, the attention mechanism of the Transformer has already been proven effective in handling multi-modal information (Nguyen et al., 2024; Zhou et al., 2023; Yu et al., 2023a; Wu et al., 2023a).

## 3 METHOD

### 3.1 OVERVIEW

**Revisiting MOTR.** The overall pipeline is illustrated in Fig. 2. We follow the general structure and basic tracking mechanism of MOTR (Zeng et al., 2022), treating MOT as an iterative sequence prediction problem. In MOTR, each trajectory is represented by a track query. Following a DETR-like structure (Carion et al., 2020), detect queries $Q_{\text{det}}^{t=1}$ for the first frame $f_{t=1}$ are fed into the Transformer decoder, where they interact with the image features $E_{\text{img}}^{t=1}$ extracted by the Transformer

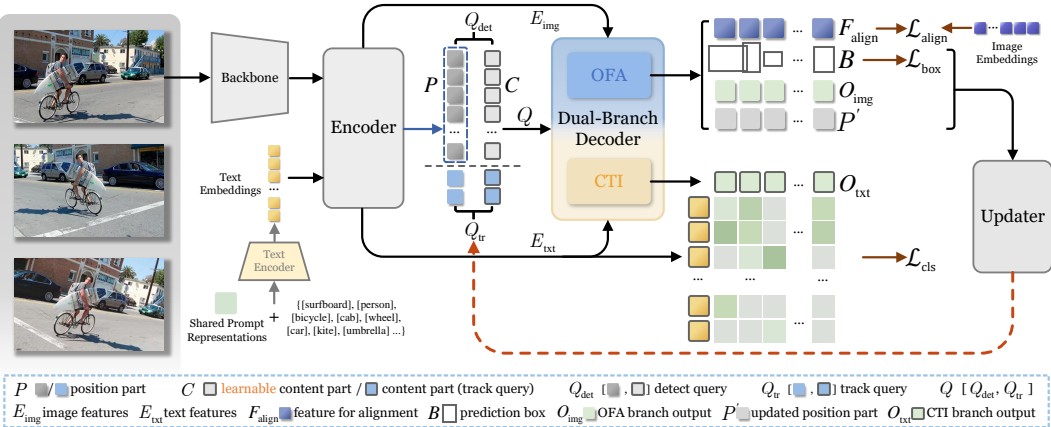

Figure 2: **Overview of OVTR.** OVTR processes two modalities of input, with modality interaction structures in both the encoder and decoder. The dual-branch decoder has the OFA branch, which serves as the medium through which the CLIP image encoder guides our model to achieve visual generalization capabilities, and the CTI branch, which handles open-vocabulary interaction and classification. The updated outputs are used as prior queries for the next frame's predictions.

encoder. This process yields updated detect queries $Q'^{t=1}_{\text{det}}$ that contain object information. Detection predictions, including bounding boxes $B^{t=1}_{\text{det}}$ and object representations $O^{t=1}_{\text{det}}$, are subsequently extracted from $Q'^{t=1}_{\text{det}}$. In contrast to DETR, for the query-based iterative tracker, $Q'^{t=1}_{\text{det}}$ are only needed to detect newly appeared objects in the current frame. Consequently, one-to-one assignment is performed through bipartite matching exclusively between $Q'^{t=1}_{\text{det}}$ and the ground truth of the newly appeared objects, rather than matching with the ground truth of all objects.

The matched $Q'^{t=1}_{\text{det}}$ will be used to update and generate the track queries $Q^{t=2}_{\text{tr}}$, which, for the second frame $f_{t=2}$, are fed once again into the Transformer decoder and interact with the image features $E^{t=2}_{\text{img}}$ to extract the representations and locations of the objects targeted by $Q^{t=2}_{\text{tr}}$, thereby enabling tracking predictions. Subsequently, the $Q^{t=2}_{\text{tr}}$ maintain their object associations and are updated to generate the $Q^{t=3}_{\text{tr}}$ for the third frame $f_{t=3}$. Parallel to $Q^{t=2}_{\text{tr}}$, and similar to the process for $f_{t=1}$, $Q^{t=2}_{\text{det}}$ are fed into the decoder to detect newly appeared objects. After binary matching, the matched $Q'^{t=2}_{\text{det}}$ are transformed into new track queries and added to $Q^{t=3}_{\text{tr}}$ for $f_{t=3}$. The entire tracking process can be extended to subsequent frames. Regarding optimization, MOTR (Zeng et al., 2022) employs multi-frame optimization, where the loss is computed by considering both ground truth and matching results. The matching results for each frame include both the maintained track associations and the binary matching results between $Q'_{\text{det}}$ and newly appeared objects.

**Tracking Mechanism During Inference.** Similar to MOTR, the network forward process during inference in OVTR follows the same procedure as during training. The key difference lies in the conversion of track queries. In detection predictions, if the category confidence score exceeds $\tau_{\text{det}}$, the corresponding updated detect query is transformed into a new track query, initiating a new track. Conversely, if a tracked object is lost in the current frame (confidence $\leq \tau_{\text{tr}}$), it is marked as an inactive track. If an inactive track is lost for $T_{\text{miss}}$ consecutive frames, it is completely removed.

**Empowering Open Vocabulary Tracking.** Leveraging the iterative nature of the query-based framework, OVTR transfers information about tracked objects across frames, aggregating category information throughout continuous image sequences to achieve robust classification performance.

In the encoder, preliminary image features from the backbone and text embeddings from the CLIP model (Radford et al., 2021) are processed through pre-fusion to generate fused image features $E_{\text{img}}$ and text features $E_{\text{txt}}$. We propose a *dual-branch decoder* comprising the OFA branch and the CTI branch. Upon input of $Q = [Q_{\text{det}}, Q_{\text{tr}}]$, the two branches respectively enable the model to gain open-vocabulary generalization capabilities in image perception and facilitate deep modality interaction with $E_{\text{txt}}$, leading to the outputs $O_{\text{img}}, O_{\text{txt}}$. $O_{\text{img}}$ serve as the input for the category information propagation (CIP) strategy, injecting category information into the *category information flow*. This process is an extension of the aforementioned mechanism where $Q'^{t}_{\text{det}}$ generates $Q^{t+1}_{\text{tr}}$. Meanwhile, $O_{\text{txt}}$ are utilized for computing category logits and for contrastive learning.

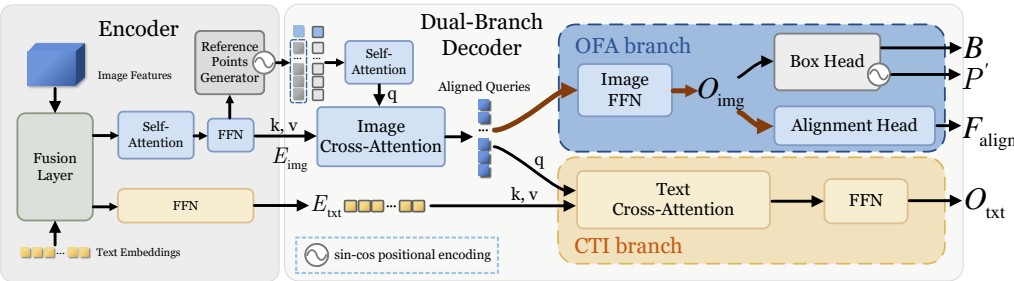

Figure 3: **Architectures of the dual-branch decoder and the encoder.** After modality fusion in the encoder, the resulting image and text features are separately fed into the decoder's Image Cross-Attention and Text Cross-Attention for interactions. Aligned queries are processed by the OFA and CTI branches to generate bounding boxes $B$, alignment features $F_{\text{align}}$, and branch outputs $O_{\text{txt}}$.

## 3.2 LEVERAGING ALIGNED QUERIES FOR SEARCH IN CROSS-ATTENTION

The perception part of OVTR builds on Zhang et al. (2022a), incorporating visual-language modality fusion in both the encoder and decoder. To efficiently conduct multimodal interaction and learn generalization ability, the decoder adopts a dual-branch structure, consisting of the object feature alignment (OFA) branch and the category text interaction (CTI) branch.

**Generating Image and Text Embeddings.** To obtain the text modality input, we feed the text and prompts (Du et al., 2022) into the CLIP (Radford et al., 2021) text encoder to generate text embeddings. Simultaneously, we use ground truth boxes to generate image embeddings via the CLIP image encoder and combine embeddings of the same category into a single representation.

Unlike the time-consuming preprocessing (Gu et al., 2021; Du et al., 2022) that generates numerous image embeddings from proposals, which are produced by an additional RPN-based detector and partially contain novel categories (unseen during training), our approach is simpler. We only need image embeddings for the closed-set categories. The embeddings are prepared offline.

**Feature Pre-fusion and Enhancement.** In the encoder, inspired by multi-modal detectors such as GLIP (Li et al., 2022a) and Grounding DINO (Liu et al., 2023), we integrated image-to-text and text-to-image cross-attention modules for feature fusion, which enhance image and text representations, preparing them for interaction in the decoder. Since the encoder outputs preliminary content features that may introduce misguidance for the decoder, we follow the approach of DINO-DETR (Zhang et al., 2022a) by generating the content parts of our queries through learnable initialization, while the position parts are derived from the reference points produced by $E_{\text{img}}$, the outputs of the encoder, through sin-cos positional encoding.

**Dual-Branch Structure.** As shown in Fig. 3, in the dual-branch structure, the OFA branch consists of a feedforward network followed by a box head and an alignment head, while the category text interaction (CTI) branch comprises text cross-attention followed by a feedforward network.

To achieve zero-shot capabilities, we utilize the OFA branch for alignment, distilling visual generalization from the CLIP image encoder and guiding the queries produced by the image cross-attention layer, referred to as *aligned queries*, to align with CLIP image embeddings. Since the CLIP image embeddings are aligned with the CLIP text embeddings and the text features $E_{\text{txt}}$ originate from the CLIP text embeddings, aligned queries that correspond to certain categories are able to align implicitly with the $E_{\text{txt}}$ of the same categories. This alignment enables the aligned queries to effectively focus on the category information conveyed in $E_{\text{txt}}$ during the text cross-attention in the CTI branch. Consequently, when images and text inputs containing novel categories are provided, the generalized network produces aligned queries that implicitly capture the features of novel categories. This enables the aligned queries to interact effectively with the $E_{\text{txt}}$ of novel categories, yielding $O_{\text{txt}}$ rich in novel category information to enhance open-vocabulary classification. The box head is incorporated into the OFA branch to ensure that the features of aligned queries are instance-level.

Specifically, We distill knowledge from the CLIP image encoder by aligning the outputs $F_{\text{align}} \in \mathbb{R}^{n \times d}$ from the alignment head with the ground truth CLIP image embeddings $V_{\text{gt}} \in \mathbb{R}^{n \times d}$. $F_{\text{align}}$

corresponds to the matching results mentioned in Sec. 3.1. Each feature is a $d$-dimensional vector, where $n$ represents the number of ground truth objects. The alignment loss $\mathcal{L}_{\text{align}}$ is formulated as:

$$\mathcal{L}_{\text{align}} = \frac{1}{n \cdot d} \sum_{i=1}^{n} \sum_{j=1}^{d} (F_{i,j,\text{align}} - V_{i,j,\text{gt}})^2, \tag{1}$$

Additionally, the dual-branch structure also aims to prevent category text information from affecting the localization ability.

### 3.3 ATTENTION ISOLATION FOR DECODER PROTECTION

For the decoder, interference may arise from both multiple category information and the content of the track queries. Specifically, interactions between queries in the self-attention layers can entangle information corresponding to multiple categories, hurting classification performance. Moreover, as a tracking-by-query framework, the decoder processes both track and detect queries in parallel. Track queries contain content about tracked objects, leading to a content gap between them and the initial detect queries. This gap may cause conflicts within the decoder layers due to the interactions in self-attention. To address these, we propose attention isolation strategies for decoder protection.

**Category Isolation Strategy.** The output features of the CTI branch $O_{\text{txt}}$, undergo dot products with the text features $E_{\text{txt}}$, followed by a softmax operation to produce the category score matrix $\boldsymbol{S} \in \mathbb{R}^{N \times M}$, where $N$ denotes the number of queries $Q$, and $M$ represents the number of selected categories. We calculate the KL (Kullback-Leibler) divergence (Kullback, 1951) between the category score distributions (vectors in $\boldsymbol{S}$) of each two predictions in $O_{\text{txt}}$ to form a matrix, called the difference matrix $\boldsymbol{D} \in \mathbb{R}^{N \times N}$. The specific formula is as follows:

$$D_{i,j} = D_{\text{KL}}(\boldsymbol{S}_{i,:} \| \boldsymbol{S}_{j,:}) + D_{\text{KL}}(\boldsymbol{S}_{j,:} \| \boldsymbol{S}_{i,:}) = \sum_{k=1}^{M} S_{i,k} \ln \left( \frac{S_{i,k}}{S_{j,k}} \right) + \sum_{k=1}^{M} S_{j,k} \ln \left( \frac{S_{j,k}}{S_{i,k}} \right), \tag{2}$$

where $D_{i,j}$ is the sum of the forward and the reverse KL divergences between the category score vectors in $\boldsymbol{S}$ corresponding respectively to the $i$-th and $j$-th queries. $D_{\text{KL}}$ represents the calculation of KL divergence.

We compute the difference matrix $\boldsymbol{D}$ based on $\boldsymbol{S}$ of the current decoder layer to generate the category isolation mask ($\boldsymbol{I} \in \mathbb{R}^{N \times N}$), which is then added to the attention weights of self-attention layer of the next decoder layer. The category isolation mask $\boldsymbol{I}$ is generated as follows:

$$I_{i,j} = \begin{cases} \text{True}, & \text{if } D_{i,j} > \tau_{\text{isol}} \\ \text{False}, & \text{if } D_{i,j} \leq \tau_{\text{isol}} \end{cases}, \tag{3}$$

where $\tau_{\text{isol}}$ is the threshold for the difference matrix $\boldsymbol{D}$.

When $I_{i,j}$ is "True", it means the category information carried by these two queries is substantially different. This difference may cause interference between the queries during self-attention. In addition, interactions among $Q_{\text{tr}}$, which have inherited robust category information, will be maintained to ensure that the self-attention mechanism inhibits duplicate predictions derived from $Q_{\text{tr}}$.

**Content Isolation Strategy.** To mitigate the impact of the content gap between track and detect queries as they jointly enter the decoder, we introduced a content isolation mask. The content distribution differs between the first frame detection and subsequent tracking, and in the latter case, track queries may interfere with detect queries. Using a vanilla decoder for both could lead to conflicts. To ensure consistent decoder operations across the two processes, we propose the content isolation mask to prevent track queries from interfering with the content of detect queries. Specifically, this mask is added to the attention weights of self-attention in the first decoder layer, with the mask positions for detect queries and track queries that attend to each other set to True to suppress their interaction. For further details on these two strategies, please refer to appendix A.4.

### 3.4 TRACKING WITH CATEGORY INFORMATION PROPAGATION ACROSS FRAMES

To enable continuous category perception and localization, we leverage the iterative nature of the query-based method and propose the category information propagation (CIP) strategy to aggregate tracked object information, thereby reinforcing category priors throughout multi-frame predictions.

Inspired by MOTR (Zeng et al., 2022), we use the a modified Transformer decoder layer for the CIP strategy. We use the outputs of the OFA branch corresponding to the matched updated queries $Q'^t_* = [P'^t_*, C^t_*]$ for the $t$-th frame $f_t$, denoted as $O^{*t}_{\text{img}}$, to update $Q'^t_*$ into the track queries $Q^{t+1}_{\text{tr}} = [P^{t+1}_{\text{tr}}, C^{t+1}_{\text{tr}}]$ for the frame $f_{t+1}$. $C^{t+1}_{\text{tr}}$ is the content part of $Q^{t+1}_{\text{tr}}$, which is the core of propagating category information between frames. It is performed in the updater of Fig. 2 and formulated as:

$$O_{\text{sa}} = \text{MHA}(O^{*t}_{\text{img}} + P'^t_*, O^{*t}_{\text{img}} + P'^t_*, O^{*t}_{\text{img}}),$$
$$O_{\text{r}} = \text{LayerNorm}(O^{*t}_{\text{img}} + \text{Dropout}(O_{\text{sa}})),$$
$$C^{t+1}_{\text{tr}} = \text{FFN}(\text{FFN}(O_{\text{r}}, O^{*t}_{\text{img}}), C^t_*),$$

(4)

where $P'^t_*$ and $C^t_*$ respectively represent the updated position parts and the content parts of $Q'^t_*$, MHA and FFN denote multi-head attention and feedforward network, and $O_{\text{sa}}$ and $O_{\text{r}}$ are the intermediate attention outputs and residual outputs. We use the sum of $O^{*t}_{\text{img}}$ and $P'^t_*$ as the queries and keys, while using $O^{*t}_{\text{img}}$ alone as the values for the MHA.

This network aggregates the category information from $O^{*t}_{\text{img}}$ with historical content $C^t_*$, providing category priors in $C^{t+1}_{\text{tr}}$ for the next frame predictions. In this way, category information is propagated to the next frame, enabling multi-frame propagation during the iterations of track queries. Meanwhile, the bounding boxes $B^t_*$ corresponding to $Q'^t_*$ are transformed via sin-cos positional encoding into $P^{t+1}_{\text{tr}}$ of $Q^{t+1}_{\text{tr}}$. We exclusively use the output representations from the OFA branch, instead of the CTI branch, as the OFA branch contains less direct textual information. This can reduce the content gap between detect and track queries. Additionally, since $O^{*t}_{\text{img}}$ is derived from the image cross-attention layer, it may more readily align with similar information within that layer when re-entering, thereby facilitating the continuous capture of the target.

### 3.5 Optimization

When an image sequence of $N$ frames is input, the multi-frame predictions are denoted as $\widehat{y} = \{\widehat{y}_i\}^N_{i=1}$, and the corresponding ground truth as $y = \{y_i\}^N_{i=1}$. We compute the loss $\mathcal{L}_{\text{seq}}$ across multiple frames, including both tracking loss and detection loss, which share the same components. The difference is that tracking loss focuses on localizing previously recognized targets, while detection loss handles newly detected ones. $\mathcal{L}_{\text{seq}}$ can be formulated as follows:

$$\mathcal{L}_{\text{seq}} = \frac{\sum^N_{n=1}(\mathcal{L}(\widehat{y}^i_{\text{tr}}\,|_{Q_{\text{tr}}}, y^i_{\text{tr}}) + \mathcal{L}(\widehat{y}^i_{\text{det}}\,|_{Q_{\text{det}}}, y^i_{\text{det}}))}{\sum^N_{n=1}(T_i)},$$

(5)

where $\widehat{y}^i_{\text{tr}}\,|_{Q_{\text{tr}}}$, $y^i_{\text{tr}}$, $\widehat{y}^i_{\text{det}}\,|_{Q_{\text{det}}}$, $y^i_{\text{det}}$ denote the association predictions, association labels, detection predictions, and unassociated labels, respectively. $T_i$ is the total number of the targets in $f_{t=i}$.

The loss function $\mathcal{L}$ includes not only the conventional classification loss and bounding box loss but also the alignment loss $\mathcal{L}_{\text{align}}$ for the OFA branch, which was mentioned in Sec. 3.2. Regarding the classification loss $\mathcal{L}_{\text{cls}}$, we perform dot products between each $O_{\text{txt}}$ and the text features $E_{\text{txt}}$ to predict logits, followed by calculating the focal loss. The single-frame loss $\mathcal{L}$ can be formulated as:

$$\mathcal{L} = \lambda_{\text{cls}}\mathcal{L}_{\text{cls}} + \lambda_{\text{L1}}\mathcal{L}_{\text{L1}} + \lambda_{\text{giou}}\mathcal{L}_{\text{giou}} + \lambda_{\text{align}}\mathcal{L}_{\text{align}},$$

(6)

where $\mathcal{L}_{\text{L1}}$ denotes the L1 loss, $\mathcal{L}_{\text{giou}}$ is the generalized IoU loss, and $\lambda_{\text{cls}}$, $\lambda_{\text{L1}}$, $\lambda_{\text{giou}}$, and $\lambda_{\text{align}}$ are the weighting parameters. We apply auxiliary losses (Al-Rfou et al., 2019) after each decoder layer.

## 4 Experiments

### 4.1 Datasets and Evaluation Metrics

**Datasets.** We conduct comparative experiments on the TAO and KITTI datasets. The TAO dataset is a diverse video tracking benchmark with 833 categories spanning various scenarios. We utilized the Open-Vocabulary MOT benchmark, based on TAO, to assess open-vocabulary tracking performance. This benchmark adopts the category division setup from prior work, treating rare classes in LVIS as

Table 1: Open-vocabulary MOT performance comparison on TAO dataset. All methods use ResNet50 (He et al., 2016) as the backbone. G-LVIS: Data generated in OVTrack (Li et al., 2023) for training association, with the same number of images as LVIS. Proposals$_{novel}$: The use of image embedding generated from proposals that partially contain novel categories for distillation. Embeds: The required number of CLIP image embeddings. †: Distills segmentation knowledge from SAM.

| Method | Elements | | | Novel | | | | Base | | | |
|---|---|---|---|---|---|---|---|---|---|---|---|
| Validation set | Data | Embeds | Proposals$_{novel}$ | TETA↑ | LocA↑ | AssocA↑ | ClsA↑ | TETA↑ | LocA↑ | AssocA↑ | ClsA↑ |
| DeepSORT (ViLD)(Wojke et al., 2017) | LVIS,TAO | 99.4M | ✓ | 21.1 | 46.4 | 14.7 | 2.3 | 26.9 | 47.1 | 15.8 | 17.7 |
| Tracktor++ (ViLD)(Bergmann et al., 2019) | LVIS,TAO | 99.4M | ✓ | 22.7 | 46.7 | 19.3 | 2.2 | 28.3 | 47.4 | 20.5 | 17.0 |
| OVTrack(Li et al., 2023) | G-LVIS,LVIS | 99.4M | ✓ | 27.8 | 48.8 | 33.6 | 1.5 | 35.5 | 49.3 | 36.9 | **20.2** |
| MASA(Li et al., 2024)† | LVIS | 99.4M | ✓ | 30.0 | 54.2 | 34.6 | 1.0 | 36.9 | 55.1 | 36.4 | 19.3 |
| OVTR | LVIS | 1,732 | | **31.4** | **54.4** | **34.5** | **5.4** | **36.6** | **52.2** | **37.6** | 20.1 |
| (vs. OVTrack) | - | - | - | (+3.6) | (+5.6) | (+0.9) | (+3.9) | (+1.1) | (+2.9) | (+0.7) | |
| Test set | Data | Embeds | Proposals$_{novel}$ | TETA↑ | LocA↑ | AssocA↑ | ClsA↑ | TETA↑ | LocA↑ | AssocA↑ | ClsA↑ |
| DeepSORT (ViLD)(Wojke et al., 2017) | LVIS,TAO | 99.4M | ✓ | 17.2 | 38.4 | 11.6 | 1.7 | 24.5 | 43.8 | 14.6 | 15.2 |
| Tracktor++ (ViLD)(Bergmann et al., 2019) | LVIS,TAO | 99.4M | ✓ | 18.0 | 39.0 | 13.4 | 1.7 | 26.0 | 44.1 | 19.0 | 14.8 |
| OVTrack(Li et al., 2023) | G-LVIS,LVIS | 99.4M | ✓ | 24.1 | 41.8 | 28.7 | 1.8 | 32.6 | 45.6 | 35.4 | **16.9** |
| OVTR | LVIS | 1,732 | | **27.1** | **47.1** | **32.1** | **2.1** | **34.5** | **51.1** | **37.5** | 14.9 |
| (vs. OVTrack) | - | - | - | (+3.0) | (+5.3) | (+3.4) | (+0.3) | (+1.9) | (+5.5) | (+2.1) | |

novel categories and frequent/common classes as base categories. It evaluates a tracker's ability to handle novel categories unseen during training, simulating real-world scenarios of identifying and tracking rare objects. The KITTI dataset, comprising 21 training and 29 test sequences, focuses on autonomous driving scenarios with diverse objects, reflecting realistic driving conditions. It serves as a benchmark for evaluating trackers' generalization across datasets. For training, we leveraged the LVIS dataset, which includes 1,203 categories, offering a rich diversity of objects for open-vocabulary learning. Under the open-vocabulary setting, these categories are divided into 866 base and 337 novel categories, enabling comprehensive model training for rare object tracking.

**Evaluation metrics.** We use TETA as the metric for evaluating open-vocabulary performance. TETA separates classification from localization and association, providing an effective measure of a model's classification ability in an open-vocabulary setting. The key metrics include localization accuracy (LocA), association accuracy (AssocA), and classification accuracy (ClsA). For generalization experiments on the KITTI dataset, we use the CLEAR-MOT metrics, such as multiple object tracking accuracy (MOTA), ID F1 score (IDF1), mostly tracked rate (MT), mostly lost rate (ML), and identity switches (IDs). Among these, MOTA and IDF1 serve as the primary metrics.

## 4.2 IMPLEMENTATION DETAILS

We use image data from LVIS (Gupta et al., 2019) for training and augment it to create image sequences. To better train this query-based tracker, we go beyond basic strategies like random flipping and cropping, incorporating advanced techniques we designed such as random occlusion and dynamic mosaic augmentation. (see appendix C for details)

To accelerate convergence, we build OVTR with a ResNet50 (He et al., 2016) backbone and apply a weight-freezing strategy. Training begins with the detection components, using a batch size of 2 for 33 epochs, a learning rate of 4e-5 that decays by a factor of 10 at the 20th epoch. Next, the dual-branch decoders and the updater are trained with a batch size of 1 for 16 epochs, starting with a learning rate of 4e-5, which decays at the 13th epoch. Multi-frame training is employed, progressively increasing the number of frames from 2 to 3, 4, and 5 at the 4th, 7th, and 14th epochs, respectively. The hyperparameter $\tau_{\text{isol}}$, the threshold for the matrix $D$, is set to a multiple of its mean value due to its variability. Training is conducted on 4 NVIDIA GeForce RTX 3090 GPUs.

## 4.3 PERFORMANCE COMPARISON ON TAO DATASET

We compare OVTR with state-of-the-art methods on the TAO validation and test sets. We evaluate OVTrack (Li et al., 2023), along with existing methods like DeepSORT (Wojke et al., 2017) and Tracktor++ (Bergmann et al., 2019) using off-the-shelf OVD (results from Li et al. (2023)), against our model. All methods are trained solely on base categories, with ResNet50 (He et al., 2016) as the backbone. Notably, all methods except OVTR utilize image embeddings that contain implicit novel categories. Additionally, OVTrack leverages DDPM (Ho et al., 2020) for data generation.

Table 2: Zero-shot domain transfer to KITTI dataset. We compare OVTR with OVTrack and CenterTrack. Our method and OVTrack are both trained using only images and undergo a zero-shot cross-domain transfer evaluation, whereas CenterTrack is trained with in-domain videos.

| Method | Data | MOTA↑ | IDF1↑ | MT↑ | ML↓ | IDs↓ | Frag↓ |
|---|---|---|---|---|---|---|---|
| *Zero-shot:* | | | | | | | |
| OVTrack(Li et al., 2023) | G-LVIS,LVIS | 69.8 | 75.6 | 62.9 | 5.8 | 594 | 307 |
| OVTR | LVIS | **71.8** | **78.3** | **64.3** | **5.4** | **378** | **169** |
| *Supervised:* | | | | | | | |
| CenterTrack(Zhou et al., 2020) | KITTI | 88.7 | 85.5 | 90.3 | 2.2 | 403 | 68 |

According to the results in Tab. 1, OVTR outperforms OVTrack on TETA of both novel and base categories across the validation and test sets. Specifically, on novel ClsA, OVTR is more than three times that of OVTrack. On the test set, OVTR outperforms OVTrack by 11.8% on AssocA for novel categories. This demonstrates OVTR has better generalization in novel category classification and tracking. Furthermore, OVTR significantly surpasses OVTrack on LocA across both the validation and test sets. These results, achieved without the use of proposals containing novel category information and with less data, confirm the effectiveness of our approach. They validate the contributions of the CIP strategy and the dual-branch structure in improving open-vocabulary tracking. Additionally, we use MASA (Li et al., 2024) results from Li et al. (2025) for comparison and find that our method performs better on novel categories, even without distilling segmentation knowledge from SAM (Kirillov et al., 2023).

### 4.4 ZERO-SHOT DOMAIN TRANSFER TO KITTI DATASET

We evaluated our method and the state-of-the-art OVTrack on the KITTI validation set, as divided by CenterTrack (Zhou et al., 2020), in the zero-shot domain transfer scenario. As KITTI only evaluates the Car and Pedestrian categories, we set the inference category range to include these two categories along with nine randomly selected additional categories to increase the challenge of category diversity. Tab. 2 shows that OVTR demonstrates better results in the Car category, surpassing OVTrack by 2.9% on MOTA and 3.6% on IDF1, while reducing IDs by 36.3%. This indicates that our model generalizes well when transferred to another dataset, further validating OVTR's adaptability to diverse autonomous driving scenarios. Due to the inconsistency between the categorization of pedestrians in the KITTI dataset and the open-vocabulary task, we present additional comparison methods for the Pedestrian category in the supplementary materials.

### 4.5 ABLATION STUDY

In this subsection, we verify the effectiveness of the model architecture and strategies through ablation studies. All models are trained on the LVIS dataset, undergoing 24 epochs of detection training followed by 15 epochs of tracking training. Due to resource constraints, the ablation studies are conducted on a subset of 40,000 images. For evaluation, we use the TAO validation dataset and designate certain base categories that were not learned during training as novel categories. We use $ClsA_b$ and $ClsA_n$ to represent ClsA of base and novel categories respectively.

**Components of the Dual-Branch Decoder.** In this part, we verify the effectiveness of the various components of the dual-branch decoder. We decoupled the OFA branch for analysis, leaving only the box head in the output section of the OFA branch. The CTI branch is essential for open-vocabulary interaction and is not analyzed separately. As reported in Tab. 3, the model with the complete dual-branch structure (row 4) outperforms the model with only the CTI branch (row 1) by 6.4% on TETA, and improves by 20.6% on $ClsA_b$. In detail, each component contributes to the performance improvement. The OFA branch (row 2) improves the model slightly on AssocA, while having an effective improvement of 10.3% on $ClsA_b$. When the alignment head and alignment loss are incorporated with the OFA branch (row 4), the model performance is further improved. These findings highlight the effectiveness of the alignment-enabled OFA branch design. In addition, when alignment is used for the CTI branch (row 3), the model achieves a significant improvement on AssocA, but the classification performance remains inferior to that of the model with a complete dual-branch structure (row 4). We analyze that aligning with image embeddings introduces conflicts in the text cross-attention, which weakens the classification performance. Therefore, it is essential to utilize both the dual-branch structure and the alignment mechanism simultaneously.

Table 3: Ablation study on decoder components. Align: the alignment head and alignment loss, CTI: the category-text interaction branch, OFA: the object feature alignment branch.

| | CTI | OFA | Align | TETA | AssocA | ClsA$_b$ | ClsA$_n$ |
|---|---|---|---|---|---|---|---|
| 1 | ✓ | ✗ | ✗ | 31.2 | 31.8 | 12.6 | 1.9 |
| 2 | ✓ | ✓ | ✗ | 31.9 | 32.0 | 13.9 | 2.1 |
| 3 | ✓ | ✗ | ✓ | 32.5 | 33.9 | 13.8 | 2.0 |
| 4 | ✓ | ✓ | ✓ | **33.2** | **34.5** | **15.2** | **2.7** |

Table 4: Ablation study on the protection strategies for the decoders. Category: the category isolation strategy, Content: the content isolation strategy.

| | Category | Content | TETA | AssocA | ClsA$_b$ | ClsA$_n$ |
|---|---|---|---|---|---|---|
| 1 | ✗ | ✗ | 32.1 | 32.8 | 14.6 | 2.3 |
| 2 | ✓ | ✗ | 32.2 | 33.0 | **15.6** | 2.5 |
| 3 | ✗ | ✓ | 32.4 | 33.6 | 14.3 | 2.5 |
| 4 | ✓ | ✓ | **33.2** | **34.5** | 15.2 | **2.7** |

Table 5: Ablation study on alignment methods. We evaluate three methods: using text embeddings, image embeddings, and the average of both embeddings for alignment.

| Alignment | TETA | AssocA | ClsA$_b$ | ClsA$_n$ |
|---|---|---|---|---|
| Text | 31.6 | 32.6 | 14.0 | 2.0 |
| Image | **33.2** | **34.5** | **15.2** | 2.7 |
| Avg | 32.3 | 33.2 | 14.1 | **3.2** |

Table 6: Ablation study on inputs for CIP $I_{CIP}$. $O_{txt}$ denotes output of the category text interaction branch, while $O_{img}$ represents output of the object feature alignment branch.

| $I_{CIP}$ | TETA | AssocA | ClsA$_b$ | ClsA$_n$ |
|---|---|---|---|---|
| $O_{txt}$ | 32.5 | 33.8 | 14.7 | 1.9 |
| $O_{img}$ | **33.2** | **34.5** | **15.2** | **2.7** |

**Decoder protection Strategies.** As shown in Tab. 4, when the category isolation strategy was applied alone (row 2), it primarily enhanced classification performance, indicating that it effectively prevents interference between different category information. The content isolation strategy applied alone (row 3) resulted in an improvement on AssocA. When both strategies were applied together (row 4), they produced a synergistic effect, resulting in a 3.4% improvement on TETA and a 5.2% increase on AssocA compared to the model without either strategy, although the classification performance was slightly suppressed compared to using only the category isolation strategy (row 2).

**Modality-Specific Embeddings for Alignment.** To explore whether aligning with image embeddings yields optimal results, we conducted an ablation comparison with three settings: alignment with text embeddings, alignment with image embeddings, and alignment with the average of both embeddings. As shown in Tab. 5, aligning with image embeddings outperformed alignment with text embeddings by 5.1% on TETA. It is evident that aligning with the average of text and image embeddings leads to improved ClsA$_n$. We believe this is because the average embeddings retain the generalization ability provided by image embeddings and enable more direct interactions due to the inclusion of text embeddings. However, using average embeddings leads to suboptimal association performance, indirectly resulting in a lower ClsA$_b$. In conclusion, aligning with image embeddings yields improved results in open-vocabulary tracking.

**Inputs for Category Information Propagation.** In this part, we aim to compare the performance of the model when using the output of OFA, $O_{img}$ versus the output from CTI, $O_{txt}$, as input for the CIP strategy. As shown in Tab. 6, using $O_{img}$ demonstrates better open-vocabulary tracking performance compared to $O_{txt}$. This suggests that the category information flow, carrying $O_{img}$ aligned with CLIP image embeddings, facilitates improved target capture and continuous classification.

## 5  CONCLUSION

In this paper, we propose OVTR, the first end-to-end open-vocabulary multiple object tracker that jointly models motion, appearance, and category. By leveraging the category information propagation strategy, we establish a higher-level category information flow, enabling the model to classify and track in a stable and continuous manner. The introduction of a dual-branch structure and protective strategies in the decoder enhances the model's generalization capability, fosters deep modality interaction, and allows for harmonious operation of classification and tracking. Our method eliminates the need for explicit track associations, complex post-processing, and time-consuming preprocessing, resulting in faster inference speeds and a simplified workflow. Despite being RPN-free and not relying on proposals containing novel categories, our model demonstrates strong generalization capabilities, achieving robust open-vocabulary tracking. As a Transformer-based framework, it is data-friendly, scalable, and has potential for further optimization. We hope this end-to-end model provides a more promising solution for open-vocabulary tracking in diverse scenarios.

## ACKNOWLEDGEMENTS

This work was supported by the National Natural Science Foundation of China under Grant 62176096.

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

## A    MORE IMPLEMENTATION DETAILS

### A.1    CATEGORY SELECTION BASED ON DISTRIBUTION.

During training, for each frame input, we randomly sample 250 categories based on the dataset's class distribution, including the ground truth categories of the current frame. These categories are represented as text embeddings, defining the scope of classes for classification learning in the current frame. Specifically, in addition to the ground truth of the current frame, the probability of selecting a certain category as a negative class is positively correlated with the 0.7th power of the number of objects in that category. This approach helps mitigate the impact of the long-tailed distribution in the dataset.

### A.2    GENERATION OF IMAGE EMBEDDINGS

We use the CLIP (Radford et al., 2021) model to generate image and text embeddings. For the generation of image embeddings, specifically, rather than using a large number of proposals detected by an additional RPN-based detector, which may include novel categories, we simply extract a specific number of cropped images for each base category from the dataset. Using the ground truth bounding boxes, we crop the images with a factor of 1.2 to create these cropped images. The cropped images for each category are then fed into the CLIP image encoder. The resulting representations are normalized and averaged to generate the image embedding for that category.

### A.3    LEARNABLE SCALING PARAMETERS

The category score matrix obtained through dot products is scaled to an appropriate magnitude before applying softmax. Learnable scaling parameters are used for scaling, with each layer having independent scaling parameters to maintain the operational independence across decoder layers.

### A.4    DETAILS OF ATTENTION ISOLATION STRATEGIES

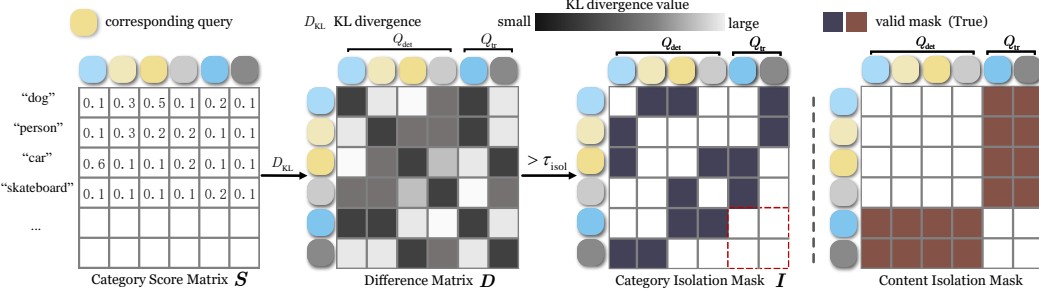

Figure 4: **Attention isolation masks.** In the difference matrix, The darker areas indicate a smaller KL divergence, meaning the category prediction distributions of the corresponding queries in the current layer are more similar. This suggests that the category information of the corresponding input queries passed to the next layer is similar. The darker areas of the masks represent masked positions, while the red dashed box shows that interactions among track queries will be maintained.

The specific masks of the category isolation strategy and the content isolation strategy are illustrated in Fig. 4. As mentioned in Sec. 3.3, the output $O_{txt}$ of the current decoder layer predicts the Category Score Matrix $S$, which is used to compute the KL divergence between category predictions for each pair of queries, resulting in the Difference Matrix $D$. $D$ captures the similarity of category information among queries. $\tau_{isol}$ is then applied to $D$ to generate an isolation mask $I$, which sets the positions with excessively large KL divergence values to True, thereby masking these positions. This mask $I$ is subsequently utilized in the self-attention layer of the next decoder layer. Since queries are updated and propagated between decoder layers, the updated queries ($[P', O_{txt}]$) output by the current layer serve as the input queries for the next layer. Therefore, the similarity captured in $D$ from the current layer are well-suited for application to the input queries of the next layer.

Specifically, $I$ is added to the attention weights obtained from the batch matrix multiplication between $q$ and $k$ (where both $q$ and $k$ are input queries) during the self-attention in decoder layers 2

Table 7: Verifying the effectiveness of components in OVTR. Isol: isolation (protection) strategies, Dual: dual-branch structure.

| | CIP | Isol | Dual | TETA | AssocA | ClsA$_b$ | ClsA$_n$ |
|---|---|---|---|---|---|---|---|
| 1 | ✗ | ✗ | ✗ | 28.9 | 29.1 | 10.7 | 1.7 |
| 2 | ✓ | ✗ | ✗ | 30.3 | 30.2 | 12.1 | 1.8 |
| 3 | ✓ | ✓ | ✗ | 31.2 | 31.8 | 12.6 | 1.9 |
| 4 | ✓ | ✗ | ✓ | 32.1 | 32.8 | 14.6 | 2.3 |
| 5 | ✓ | ✓ | ✓ | **33.2** | **34.5** | **15.2** | **2.7** |

Table 8: Different image sequence lengths for multi-frame optimization. The maximum image sequence lengths set for the four experiments are 2, 3, 4 and 5 respectively.

| Length | TETA | AssocA | ClsA$_b$ | ClsA$_n$ |
|---|---|---|---|---|
| 2 | 27.9 | 24.9 | 13.5 | 2.2 |
| 3 | 30.5 | 32.2 | 14.5 | 2.5 |
| 4 | 31.6 | 33.8 | 14.7 | 2.4 |
| 5 | **33.2** | **34.5** | **15.2** | **2.7** |

to 6. Because the coordinates $(i, j)$ of $I$ corresponding to queries $i$ and $j$ with significant category information differences are set to True (in practice, the value is set to $-\infty$), the attention weight at this position becomes 0 after the softmax operation. This enforces attention isolation, preventing category information from interfering with each other. It is worth noting that this mask is not used in the first decoder layer, as the content part $C$ of the queries fed into the self-attention in the first decoder layer is initialized through learnable parameters and does not contain category information.

The implementation of the content isolation mask is straightforward, as it simply sets the positions for detect queries and track queries that attend to each other to True. This mask is applied solely to the self-attention layer of the first decoder layer for two reasons: first, the content gap between $Q_{\text{det}}$ and $Q_{\text{tr}}$ diminishes after the first decoder layer interaction, as detection queries carry enough object semantic content; and second, to ensure that track queries can capture global information. Apart from being applied in different decoder layers, the content isolation mask performs the same operation in self-attention as the previously mentioned category isolation mask, applied to the attention weights to restrict the interaction between queries.

# B ADDITIONAL EXPERIMENTS

## B.1 VERIFYING THE EFFECTIVENESS OF COMPONENTS IN OVTR

In this part, we verify the effectiveness of three core components in OVTR: the category information propagation (CIP) strategy, the dual-branch structure, and the decoder isolation protection strategies. The row 1 represents the baseline model we constructed, which does not incorporate category information when iterating track queries, essentially lacking a category information flow. Specifically, in the single-branch structure (CTI only), we designed an auxiliary network structure identical to the standard MOTR (Zeng et al., 2022) decoder, but it operates entirely independently of the CTI branch. As a result, the queries passed through this structure lack category information. When using its output as input to CIP, the category information flow lacks direct category information.

The results are reported in Tab. 7. According to the results, all the components have boosted the open-vocabulary tracking performance effectively. OVTR (row 5) achieves a 14.9% improvement on TETA compared to the constructed baseline (row 1). The adoption of the CIP strategy increases AssocA by 3.8% and ClsA$_b$ by 13.1%, significantly improving both association and classification. This suggests that the inclusion of CIP enables better collaboration between classification and tracking. Specifically, with the addition of the CIP strategy, the dual-branch structure (row 4) provides the greatest performance gain. Compared to using CIP alone (row 2), TETA increases by 5.9%, and both AssocA and ClsA see significant improvements, particularly with ClsA$_b$ increasing by 20.7%. This suggests that the designed dual-branch structure enhances generalization performance and modality interaction, leading to a powerful open vocabulary capability. This also demonstrates that the dual-branch structure strengthens tracking effectively by feeding the aligned representations into the category information flow, helping capture objects and enhance tracking. In contrast, the isolation strategies (row 3) provide a smaller gain compared to the dual-branch structure, but still improve AssocA by 5.3% compared to using CIP alone. This indicates that the protection provided to the decoder is effective, helping maintain continuity between the initial frame detection and subsequent tracking, thus improving tracking performance.

## B.2 ANALYSIS OF IMAGE SEQUENCES LENGTH FOR OPTIMIZATION

To investigate whether multi-frame optimization helps the model learn more stable category information transfer and tracking, we set the maximum number of optimization frames in the ablation

Table 9: Open-vocabulary MOT inference speed test on TAO dataset. OVTR-Lite excludes the category isolation strategy and tensor KL divergence computation.

| Method | Speed | Novel | | | | Base | | | |
|---|---|---|---|---|---|---|---|---|---|
| Validation set | FPS | TETA↑ | LocA↑ | AssocA↑ | ClsA↑ | TETA↑ | LocA↑ | AssocA↑ | ClsA↑ |
| OVTrack (Li et al., 2023) | 3.1 | 27.8 | 48.8 | 33.6 | 1.5 | 35.5 | 49.3 | 36.9 | **20.2** |
| OVTR (**Ours**) | **3.4** | **31.4** | **54.4** | **34.5** | **5.4** | **36.6** | **52.2** | **37.6** | 20.1 |
| OVTR-Lite (**Ours**) | **12.4** | 30.1 | 52.7 | 34.4 | 3.1 | 35.6 | 51.3 | 37.0 | 18.6 |

experiments to 2, 3, 4 and 5. The first experiment maintains a frame count of 2 throughout. In the second experiment, we start with 2 frames and increase the number of frames by 1 at the 4th epoch. The third experiment starts with 2 frames and increases the frame count by 1 at both the 4th and 7th epochs. The fourth experiment also starts with 2 frames but increases the frame count by 1 at the 4th, 7th, and 14th epochs. As shown in Tab. 8, when the length of the video clip gradually increases from 2 to 5, the TETA, AssocA, and $ClsA_b$ metrics improve by 19.0%, 38.6%, and 12.6%, respectively. This indicates that multi-frame joint optimization contributes to improved tracking performance and classification stability.

## B.3 INFERENCE SPEED EVALUATION

We evaluated the inference speed of OVTrack and OVTR on a single NVIDIA GeForce RTX 3090 GPU. The results reported in Tab. 9, indicate that OVTR achieves faster inference compared to OVTrack. Additionally, we tested a lightweight version, OVTR-Lite, which excludes the category isolation strategy and tensor KL divergence computation. Despite some performance trade-offs, OVTR-Lite still outperforms OVTrack in overall performance. It achieves 4 times faster inference speed compared to OVTrack, while keeping memory usage below 4GB during inference. The speed test is conducted on the TAO validation set.

## B.4 EVALUATION OF PEDESTRIAN CATEGORY ON THE KITTI DATASET

Table 10: Zero-shot domain transfer to KITTI dataset (Pedestrian category).

| Method | Data | MOTA↑ | IDF1↑ | IDs↓ | MOTP↑ | IDR↑ | FN↓ |
|---|---|---|---|---|---|---|---|
| OVTrack | G-LVIS,LVIS | 4.5 | 11.2 | 113 | 67.9 | 6.2 | 4083 |
| OVTR | LVIS | 40.5 | 56.1 | 176 | **74.1** | **56.1** | **1332** |

Since KITTI differentiates between Pedestrian and Cyclist, which the open-vocabulary task does not, this differentiation results in a higher false positive rate when evaluating the Pedestrian category, making MOTA metrics less indicative of actual performance for pedestrian tracking. To provide a more accurate assessment of pedestrian tracking, we selected IDR, FN, and MOTP to compare the model's performance on pedestrians. Although there may be numerous false positives for cyclists, comparing the recall rate for pedestrians is reasonable. The results in Tab. 10 indicate that our method significantly outperforms OVTrack in capturing and tracking pedestrian targets when transferred to the KITTI dataset. This finding is consistent with our tests on TAO, where OVTrack often fails to identify pedestrians. In addition to the model's design and performance, this may also be related to the fact that OVTrack includes pedestrians in the classification learning scope during every frame of training, while the annotations for pedestrians in LVIS are not fully comprehensive.

## B.5 EVALUATION ON DATASET WITH HIGHER ANNOTATION FRAME RATE

Table 11: Open-vocabulary MOT performance comparison on OVT-B dataset.

| Method | | Elements | | Novel | | | | Base | | | |
|---|---|---|---|---|---|---|---|---|---|---|---|
| OVT-B | Data | Embeds | Proposals_{novel} | TETA↑ | LocA↑ | AssocA↑ | ClsA↑ | TETA↑ | LocA↑ | AssocA↑ | ClsA↑ |
| OVTrack(Li et al., 2023) | G-LVIS,LVIS | 99.4M | ✓ | 45.5 | 61.1 | 65.5 | **9.6** | 46.8 | 60.5 | 66.7 | **13.4** |
| OVTrack+(Liang & Han) | G-LVIS,LVIS | 99.4M | ✓ | **46.4** | **62.5** | 67.3 | 9.4 | **47.6** | **61.6** | 68.2 | 13.2 |
| OVTR | LVIS | 1,732 | | 45.5 | 59.7 | **67.6** | 9.3 | **47.6** | 60.9 | **68.9** | 12.9 |

We compare open-vocabulary MOT performance on OVT-B dataset (Liang & Han). Tab. 11 shows that OVTR outperforms OVTrack on base TETA and matches OVTrack+ performance. Without using proposals containing novel categories, OVTR still achieves comparable novel TETA results

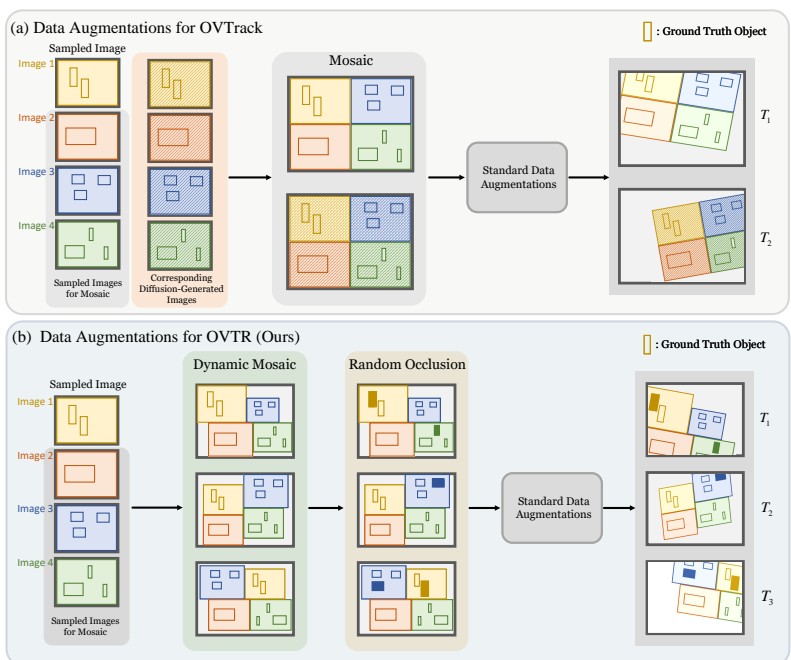

Figure 5: **OVTR data augmentations.** Unlike OVTrack, which is based on appearance matching, our method does not utilize diffusion models for data augmentation. Instead, we propose Dynamic Mosaic and Random Occlusion data augmentation to simulate object appearance and disappearance, tracking continuity after occlusion, and maintaining correct associations when relative motion occurs between tracked objects and others.

to OVTrack. Our method also achieves the highest AssocA for both base and novel categories, demonstrating superior tracking performance in high-frame-rate videos.

## C  STATIC IMAGE AUGMENTATIONS FOR QUERY-BASED TRACKING

Our multi-frame training data augmentations are summarized in Fig. 5. As a MOTR-like query-based tracker, OVTR requires a different approach to data augmentation and has higher demands for training data. Our data augmentation includes conventional techniques, such as applying random resizing, horizontal flipping, color jittering, and random affine transformations to single images to create distinguishable multi-frame data. This part aligns with OVTrack (Li et al., 2023). Additionally, we propose Dynamic Mosaic and Random Occlusion augmentations, specifically designed for MOTR-like trackers.

Unlike appearance-matching-based methods, our approach does not rely on diffusion models for additional data augmentation. Instead, it focuses on enhancing the motion realism of static images during data augmentation, making them more representative of the physical world. Specifically, while query-based trackers excel at maintaining associations over extended periods, they place higher demands on the realism of object motion patterns in training data. For OVTR, track queries must not only learn to capture the same object as it moves to a new position in the next frame, but also handle scenarios such as object appearance and disappearance, tracking continuity after occlusion, and maintaining correct associations when relative motion occurs between tracked objects and others .

To address these challenges, we propose Dynamic Mosaic augmentation, an improvement over the Mosaic augmentation in OVTrack. In addition to stitching four different images into a single composite, Dynamic Mosaic generates images with varying relative spatial relationships among objects across different training frames. This simulates scenarios such as objects approaching or receding from each other, crossing paths, and exhibiting relative size changes. The Random Occlusion augmentation is employed to simulate situations where objects disappear due to occlusion and then reappear or suddenly emerge in the scene.

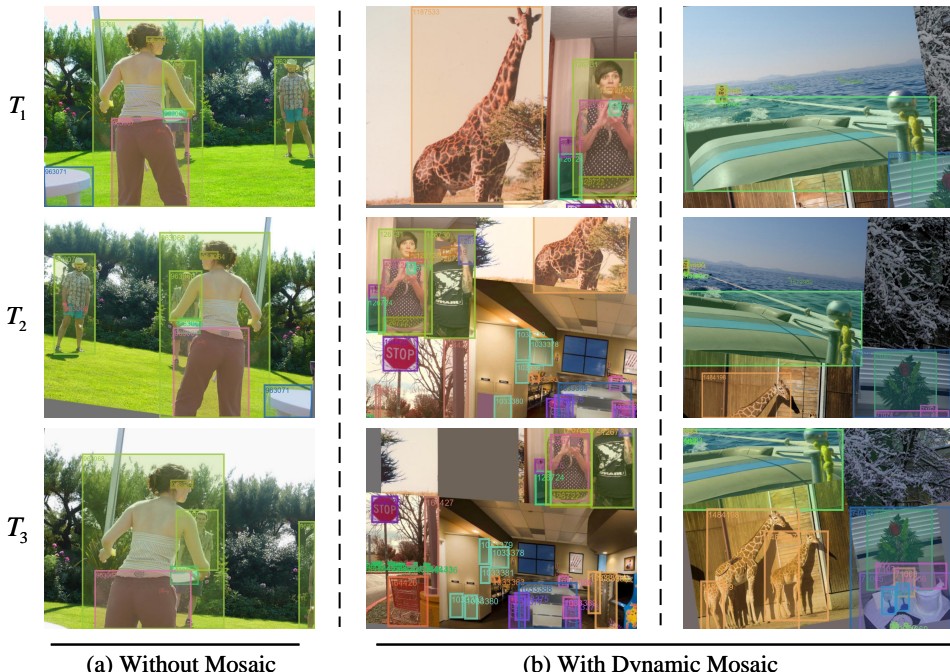

|  | (a) Without Mosaic | (b) With Dynamic Mosaic |
|---|---|---|

Figure 6: **OVTR data augmentations visualization.** From the images, it can be observed that Dynamic Mosaic augmentation introduces relative motion and relative size changes between targets, while the positional swapping of images simulates target crossing paths. Additionally, the bottom-center image illustrates a random occlusion applied to a giraffe.

Here, we detail the specific operations during training. Dynamic Mosaic is not applied to every sampled image. To avoid neglecting learning for simple scenes or larger targets, we set the probability of applying Dynamic Mosaic augmentation to a sampled training image at 0.5. In contrast, Random Occlusion is applied to every sampled image. For Dynamic Mosaic augmentation, four sampled images are processed with relative size adjustments, vertical or horizontal translations, random swaps of positions between two images, and possible horizontal flipping of one or more images. Random Occlusion requires a simple preprocessing step, where a script is used to mark targets that rarely overlap with others, based on the ground truth bounding boxes in the annotations. During Random Occlusion augmentation, only these marked targets are randomly processed, preventing unintended occlusion of other targets and avoiding negative impacts on model learning. The output images after data augmentation used for training OVTR, including cases with and without Dynamic Mosaic, are shown in Fig. 6.

## D   MODEL AND TRAINING HYPERPARAMETERS

Tab. 12 and Tab. 13 present the hyperparameters used in the detection training phase and the tracking training phase, respectively. The hyperparameters used in the tracking training phase are listed in Tab. 13, while other unmentioned parameters, such as structural parameters and loss weights, are the same as those in the detection phase. Our model follows the standard 6-encoder, 6-decoder structure in DETRs. For the update and propagation of queries between decoder layers, we specifically use the updated position part $P'$ as the input position part of the queries for the next decoder layer, while the representation $O_{\text{txt}}$ output by the CTI branch is used as the content part for the next decoder layer. This is because the CTI branch includes an extra cross-attention layer compared to the OFA branch, allowing $O_{\text{txt}}$ to contain more refined category information, leading to more accurate priors for classification in subsequent layers. The shuffle ratio, dislocation ratio, single ratio range, and occlusion ratio range are hyperparameters in the Dynamic Mosaic and Random Occlusion augmentations used to control the extent of augmentation. Sampler lengths specifies the number of frames for multi-frame training during each of the four phases. Sampler steps indicates the epochs where these transitions to different multi-frame training lengths occur.

Table 12: Hyper-parameters used in the detection training phase.

| Item | Value |
|---|---|
| optimizer | AdamW |
| lr | 4e-5 |
| weight decay | 1e-4 |
| clip max norm | 0.1 |
| number of encoder layers | 6 |
| number of decoder layers | 6 |
| dim feedforward | 2048 |
| hidden dim | 256 |
| dropout | 0.0 |
| nheads | 8 |
| number of queries | 900 |
| set cost class | 3.0 |
| set cost bbox | 5.0 |
| set cost giou | 2.0 |
| ce loss coef | 2.0 |
| bbox loss coef | 5.0 |
| giou loss coef | 2.0 |
| alignment loss coef | 2.0 |

Table 13: Additional hyper-parameters used in the tracking training phase.

| Item | Value |
|---|---|
| lr | 4e-5 |
| lr of backbone | 4e-6 |
| sampler steps | 4, 7, 14 |
| sampler lengths | 2, 3, 4, 5 |
| shuffle ratio | 0.1 |
| dislocation ratio | 0.25 |
| single ratio range | 0.7, 1.2 |
| occlusion ratio range | 0.1, 0.13 |

## E  VISUALIZATION

As shown in the figures, the results on the left represent the tracking outcomes of OVTR, while the results on the right depict those of OVTrack. Overall, it can be observed from the four sets of images that our method experiences fewer ID switches and results in fewer false positives.

In Fig. 7, it can be seen that our tracker, OVTR (on the left), maintains stable tracking of the three sheep without any ID switches, whereas OVTrack (on the right) experiences ID switches and loses many previously detected targets. In Fig. 8, OVTrack (on the right) generates a significantly higher number of false positives compared to our method. In Fig. 9, on the left, our OVTR tracks both the person and the components of the clothing and skateboard, where the person (ID 5) and the jacket (ID 2) remain consistently tracked over 15 frames (after sampling every 30 frames) in a high-speed scenario. The tracking remains stable, and the correct categories are preserved, demonstrating the effectiveness of our dual-branch structure and CIP strategy. In contrast, OVTrack reaches ID 110 at Frame 17, indicating a large number of ID switches, suggesting some challenges in tracking stability. In Fig. 10, in the second frame, the squirrel on the left partially occludes the squirrel on the right. After the occlusion ends in the third frame, our tracker (on the left) maintains the same ID for the squirrel as in the first frame, while OVTrack (on the right) experiences another ID switch and a significant number of classification errors.

Overall, from the tracking results, it is evident that tracking diverse targets in scenes with a variety of categories presents challenges. However, our method, combining the CIP strategy with a dual-branch structure and decoder protection strategies, achieves relatively robust tracking. The results demonstrated by OVTR suggest that it holds potential for effective tracking in such scenarios.

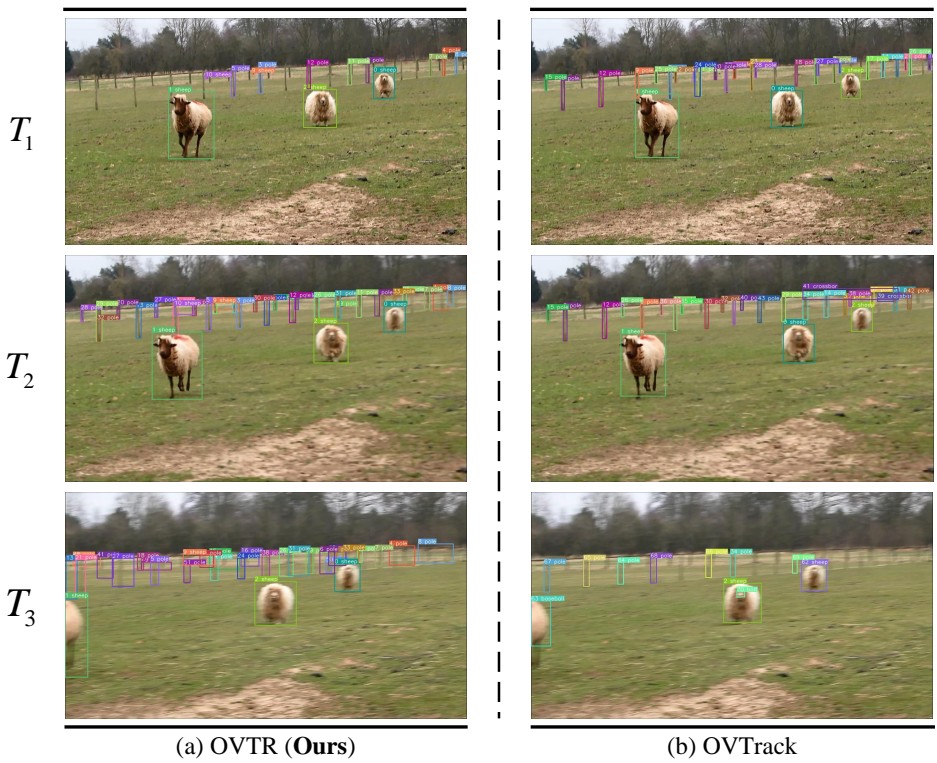

Figure 7: **Qualitative comparison of OVTR and OVTrack in multi-object motion scenario 1.**

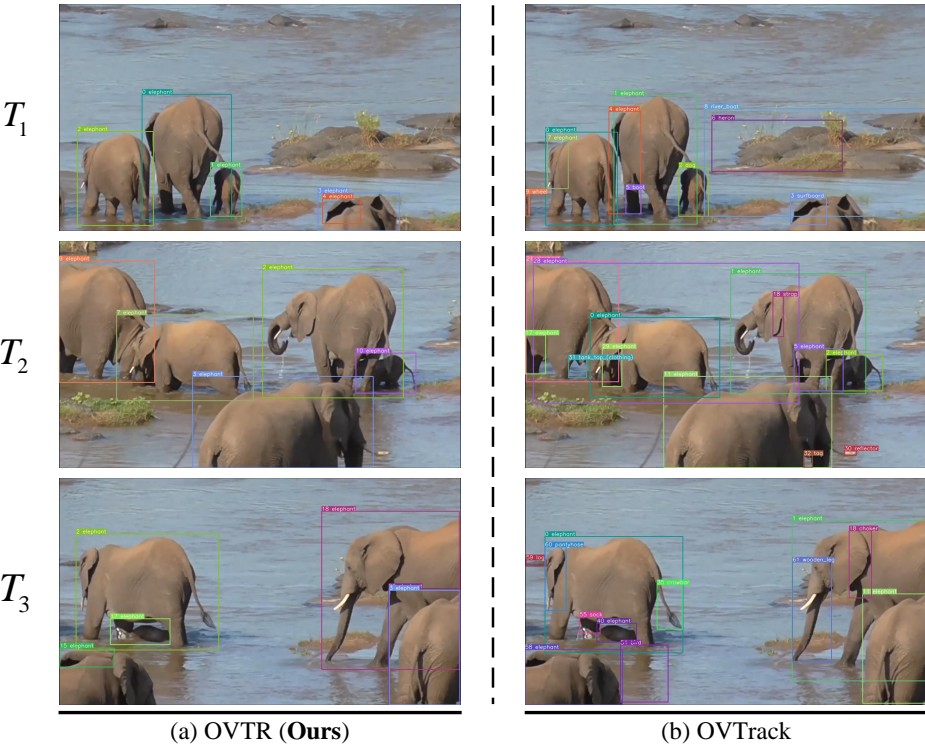

Figure 8: **Qualitative comparison of OVTR and OVTrack in multi-object motion scenario 2.**

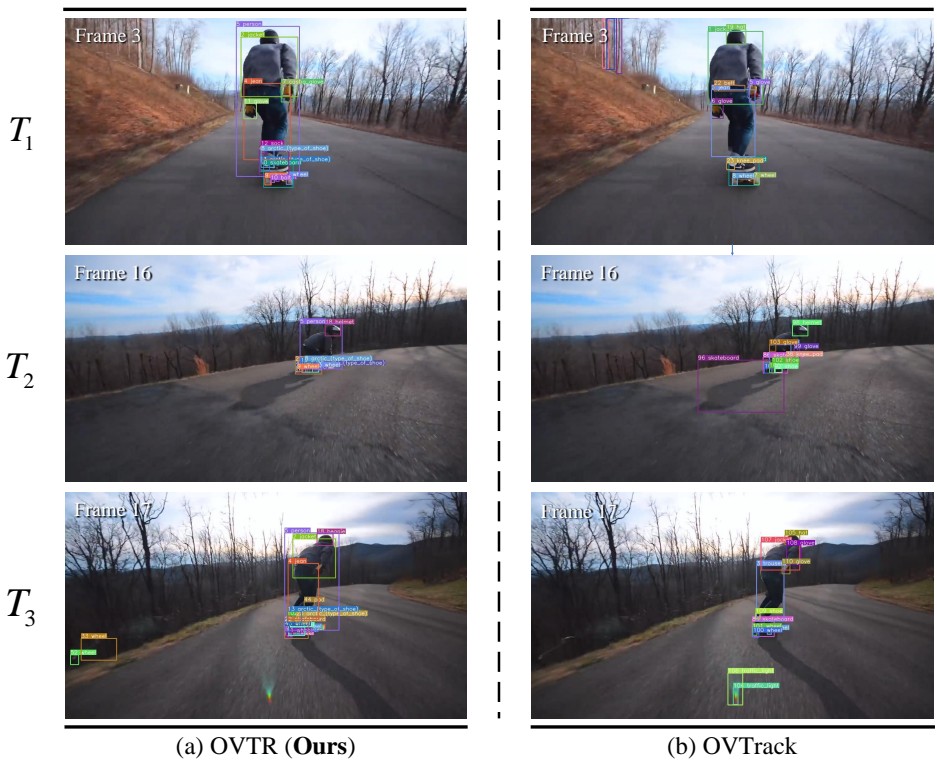

Figure 9: **Qualitative comparison in a concentrated fast-moving multi-object scenario.**

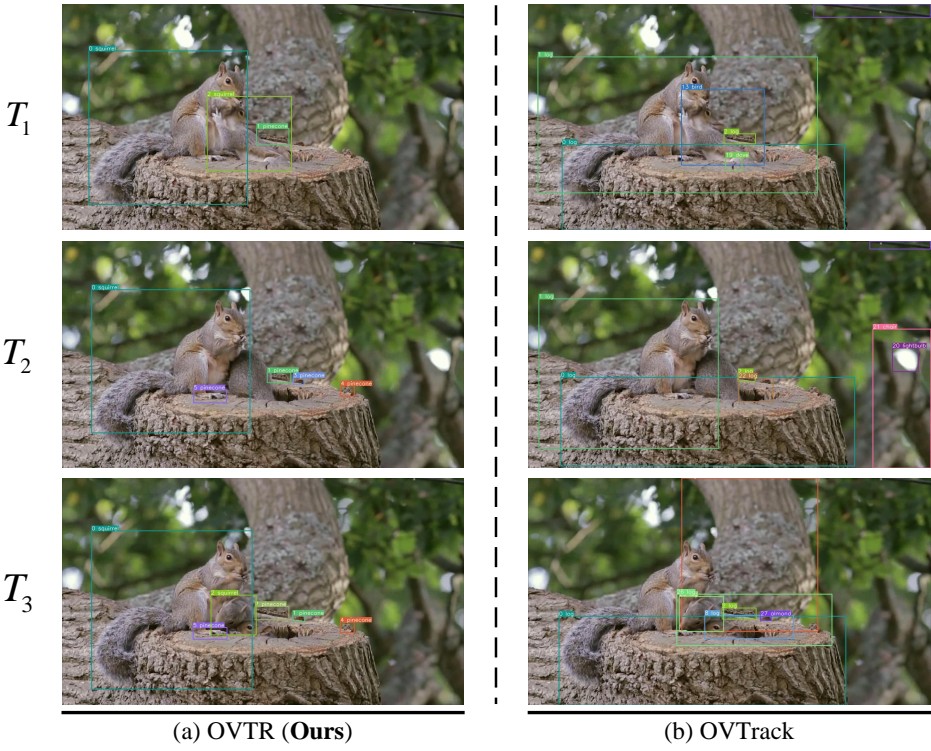

Figure 10: **Qualitative comparison on rare category objects in an occlusion scenario.**

