# OpenReview forum: "OVTR: End-to-End Open-Vocabulary Multiple Object Tracking with Transformer"
_ICLR.cc/2025/Conference — ICLR 2025 Poster_

### Official Review · Reviewer_xFtV · 2024-11-02

**Soundness:** 2
**Presentation:** 2
**Contribution:** 3
**Rating:** 6
**Confidence:** 3

**Summary:**

This paper introduces an End-to-End Open-Vocabulary Multiple Object Tracking with Transformer, OVTR, a novel approach to multi-object tracking (MOT) that extends to handle unseen categories without prior training on those specific categories. The experimental results demonstrate that OVTR outperforms existing state-of-the-art methods in tracking novel categories on standard benchmarks like TAO and KITTI, showcasing its effectiveness and adaptability.

**Strengths:**

1. End-to-End Architecture: OVTR is the first to propose an end-to-end trainable model for open-vocabulary MOT, which integrates motion, appearance, and category information simultaneously.
2. The introduction of the CIP strategy and the dual-branch structure represent creative combinations of existing ideas in a novel application domain.
3. The technical depth and rigorous experimentation by comparing OVTR with state-of-the-art methods provide a strong empirical basis for the claims made.

**Weaknesses:**

1. For the experimental results, although OVTR proposed for handling open-vocabulary multi-object tracking tasks demonstrates innovation and advancement, the performance did not meet the high expectations, particularly in terms of localization accuracy.
2. The authors should introduce more implementation details and release the source code for reproducibility.
3. The introduction of a dual-branch structure and multiple attention mechanisms increases the model complexity significantly. This might limit the practical deployment on resource-constrained platforms.

**Questions:**

1. The end-to-end MOT framework has been explored in MOT task, so what's the main difference with previous methods (such as TransTrack, TrackFormer, MOTR ...)
2. Could the authors provide visual evidence to support the effectiveness of OVTR?  Specifically, it would be beneficial to show comparative visualizations of the tracking process, including scenarios where the model successfully handles rapid motion changes, partial occlusions, and diverse object appearances across different environments.
2. What about the computational efficiency and real-time applicability of the OVTR model? Discussing its performance in terms of tracking speed and resource usage would greatly enhance the understanding of its practical value.

---

> ### Author Response · Authors · 2024-11-27
> **Response to Reviewer xFtV (Part 1)**
>
> We sincerely appreciate your efforts in reviewing our paper.
>
> - **Q1. The performance did not meet the high expectations, particularly in terms of localization accuracy.**
>
>   Thanks for your valuable comment. After submission, we further iterated and optimized our existing training strategy, **resulting in improved performance**. Specifically, the number of training epochs was increased from 12 to **15**. At the 13th epoch, the learning rate underwent an additional decay by a factor of 10. Multi-frame training was adjusted, increasing the number of frames from 2 to 3, 4, and **5** at the 4th, 7th, and **14th** epochs, respectively (previously 2, 3, and 4 at the 4th and 7th epochs). This allowed the model to accurately locate and track objects over longer video sequences. The performance of the model trained with this strategy is presented in the table below.
>
>   The results demonstrate that, compared to OVTrack, our method achieved a **12.9%** improvement on the TETA metric for novel categories and a **3.1%** improvement for base categories on the validation set. On the test set, TETA increased by **12.4%** for novel categories and **5.8%** for base categories.
>   **For localization accuracy (LocA)**, the validation set showed an improvement of **11.5%** for novel categories and **5.9%** for base categories, while the test set showed an increase of **12.7%** and **12.1%** for novel and base categories, respectively.
>
>   | **Validation set** | Data         | ${\text{Proposals}}_{\text{novel}}$ | TETA↑(novel)  | LocA↑(novel)  | AssocA↑(novel) | ClsA↑(novel) | TETA↑(base)   | LocA↑(base)   | AssocA↑(base) | ClsA↑(base)   |
>   | -- | -- | --- | -- | -- | --- | --- | -- | -- | -- | -- |
>   | OVTrack                  | G-LVIS, LVIS | ✓                                    | 27.8           | 48.8           | 33.6            | 1.5           | 35.5           | 49.3           | 36.9           | **20.2** |
>   | OVTR                     | LVIS         |                                       | **31.4** | **54.4** | **34.5**  | **5.4** | **36.6** | **52.2** | **37.6** | 20.1           |
>
>   | **Test set** | Data         | ${\text{Proposals}}_{\text{novel}}$ | TETA↑(novel)  | LocA↑(novel)  | AssocA↑(novel) | ClsA↑(novel) | TETA↑(base)   | LocA↑(base)   | AssocA↑(base) | ClsA↑(base)   |
>   | -- | -- | --- | -- | -- | --- | --- | -- | -- | -- | -- |
>   | OVTrack            | G-LVIS, LVIS | ✓                                    | 24.1           | 41.8           | 28.7            | 1.8           | 32.6           | 45.6           | 35.4           | **16.9** |
>   | OVTR               | LVIS         |                                       | **27.1** | **47.1** | **32.1**  | **2.1** | **34.5** | **51.1** | **37.5** | 14.9           |
>
>
> 	At last, we would like to emphasize that OVTR is a novel paradigm of open-vocabulary MOT that achieves **OV tracking in a fully end-to-end manner.** The motivation behind this work is to establish a novel tracking paradigm distinct from the mainstream OVD-based tracking architectures. This stems from the fact that, while existing OVD-based architectures demonstrate decent tracking performance, they also have several inherent limitations. These include the need for an additional OV-Detector, complex post-tracking processing, and a lack of scalability to larger datasets, which collectively hinder further improvements and real-world applications.
>
> 	In contrast, our proposed OVTR exhibits several advantageous characteristics: **a single model, fully end-to-end processing, no post-processing, and a data-friendly nature that facilitates scalability**. It has already demonstrated immense potential and offers the OV-MOT community a novel research direction to explore.
>
>   Additionally, our method does not require a separate RPN-based detector to pre-extract proposals (which may contain novel class information, but our method does not rely on it). Furthermore, it does not require using the CLIP image encoder to generate embeddings for proposals, which can result in hundreds of gigabytes of embeddings. Our approach **significantly saves time and reduces operations during the offline preprocessing stage.**
>
>
> - **Q2. Introduce more implementation details and release the source code for reproducibility.**
>
>   Thanks for your valuable suggestions.
>   **Implementation Details:** In the rebuttal revision, we have added supplementary materials to provide detailed information about the experiments, including specifics of the data augmentations and the hyperparameters used during training. **Details can be found in Appendix C and D of the rebuttal revision, beginning from line 805.**
>   **Code Availability:** After ICLR, we will upload the revised version of the paper to arXiv and release the code.

---

> ### Author Response · Authors · 2024-11-27
> **Response to Reviewer xFtV (Part 2)**
>
> (Continued from Part 1)
>
> - **Q3. The introduction of a dual-branch structure and multiple attention mechanisms increases the model complexity significantly, which may limit the practical deployment on resource-constrained platforms.**
>
>   **On the Dual-Branch Structure:**
>   First, **the OFA branch** within the dual-branch structure **is lightweight**, involving minimal parameter overhead, as its primary role is to guide the two attention layers preceding the branches.
>   Second, the dual-branch structure, combined with multi-head attention, **plays a key role in achieving open-vocabulary tasks**. Specifically, it equips the model with open-vocabulary generalization capabilities and enables deep modality interactions. Additionally, the inclusion of the dual-branch structure eliminates the need for pre-generated proposals from category-agnostic detectors. These proposals, when processed through CLIP, typically produce embeddings amounting to hundreds of gigabytes for knowledge distillation. By avoiding this dependency, the dual-branch structure significantly reduces preprocessing overhead.
>
>   **On the Multi-Head Attention Mechanism:**
>   The multi-head attention mechanism is essential for OVTR’s Transformer-based framework, as it supports query-based set sequence prediction. It facilitates mutual attention among queries, cross-modal attention between queries and features, and the aggregation of category information through the CIP strategy. This structure is fundamental to the model's functionality.
>
>   **Inference Speed Evaluation:**
>   We evaluated the inference speed of OVTrack and OVTR on a single NVIDIA GeForce RTX 3090 GPU. The results, summarized in the table below, indicate that OVTR achieves faster inference compared to OVTrack. Additionally, we tested a lightweight version, OVTR-Lite, which excludes the category isolation strategy and tensor KL divergence computation. Despite some performance trade-offs, OVTR-Lite still outperforms OVTrack in overall performance. It achieves nearly **4× faster inference speed** while maintaining **GPU memory usage within 4GB** during inference.
>
>   | **Validation set** | **FPS**  | TETA↑(novel)  | LocA↑(novel)  | AssocA↑(novel) | ClsA↑(novel) | TETA↑(base)   | LocA↑(base)   | AssocA↑(base) | ClsA↑(base)    |
>   | -- | -- | -- | -- | -- | --- | -- | -- | -- | -- |
>   | OVTrack                  | 3.1            | 27.8           | 48.8           | 33.6           | 1.5           | 35.5           | 49.3           | 36.9           | **20.2** |
>   | OVTR                     | **3.4**  | **31.4** | **54.4** | **34.5** | **5.4** | **36.6** | **52.2** | **37.6** | 20.1           |
>   | **OVTR-Lite**      | **12.4** | 30.1           | 52.7           | 34.4           | 3.1           | 35.6           | 51.3           | 37.0           | 18.6           |
>
>   In summary, despite being transformer-based, OVTR achieves relatively fast inference speed due to its end-to-end nature, which eliminates the need for complex post-processing and explicit similarity associations between frames.

---

> ### Author Response · Authors · 2024-11-27
> **Response to Reviewer xFtV (Part 3)**
>
> (Continued from Part 2)
>
> - **Q4. The end-to-end MOT framework has been explored in MOT task, so what's the main difference with previous methods (such as TransTrack, TrackFormer, MOTR ...)**
>
>   TransTrack and TrackFormer are transformer-based methods, but they are not strictly end-to-end approaches. While MOTR is an end-to-end method, it is designed primarily for traditional multi-object tracking scenarios, focusing on pedestrian tracking, and can only track a limited number of pre-trained categories. This makes it difficult to apply in more diverse scenes. In contrast, our open-vocabulary, end-to-end tracker, OVTR, addresses the problem of multi-object tracking in open-vocabulary and diverse scenarios. It extends the range of object categories to at least a thousand and can recognize categories unseen during training. This ability to track even unseen categories (i.e., those not encountered during training) in diverse scenarios aligns with the trend of perceptual intelligence evolving towards open-world recognition.
> - **Q5. Provide visual evidence to support the effectiveness of OVTR including scenarios where the model successfully handles rapid motion changes, partial occlusions, and diverse object appearances across different environments.**
>
>   Thanks for your valuable comments. In the rebuttal revision supplementary materials, we have updated visualizations of tracking performance in scenarios with rapid motion changes (Figure 9) and partial target occlusion (Figure 10). Together with the previous visualizations, these provide results across multiple scenarios and for objects of various categories. Please refer to **Appendix E** of the rebuttal revision, beginning from **line 912**, for more details.
> - **Q6. The computational efficiency and real-time applicability of the OVTR model?**
>
>   The discussion on tracking speed and resource usage is elaborated in **Q3**. Please refer to **Q3** for more details. Overall, our model demonstrates practical value in diverse scenarios.

---

> > ### Comment · Reviewer_xFtV · 2024-12-02
> > **Feedback to Author Responses**
> >
> > The rebuttal has addressed some of my concerns, and the current rebuttal revision is much better than the orginial submitted version. Therefore, I tend to raise my score.
> >
> > In addition, I suggest the authors release their code for reproducibility.

---

> > > ### Author Response · Authors · 2024-12-03
> > >
> > > We will promptly organize the code files and make them publicly available after ICLR.
> > >
> > > Once again, we sincerely appreciate the reviewer for time and effort in reviewing our submission.

---

### Official Review · Reviewer_3Y3k · 2024-11-03

**Soundness:** 1
**Presentation:** 2
**Contribution:** 2
**Rating:** 3
**Confidence:** 5

**Summary:**

The authors present the first application of the Transformer architecture in Open-Vocabulary Multiple Object Tracking. This work simultaneously considers the motion, appearance, and category features of the targets. A novel approach for constructing inter-frame category information flow is proposed, along with a Dual-Branch structure. As a result, an improvement of approximately one point is achieved on the Novel TETA metric of the TAO dataset.

**Strengths:**

The advantages of this work include the following:

1. It effectively leverages the strengths of the Transformer architecture and strives to apply it to the Open-Vocabulary Multiple Object Tracking (OVMOT) task.
2. The motivation throughout the paper is well-founded, as it integrates motion, appearance, and category features, utilizing category information flow to guide subsequent tracking.
3. The use of a dual-branch structure addresses the issue of increased training data arising from proposals with unknown categories during training. Instead, it employs ground truth to construct supervision, thereby reducing burdensome in training process.

**Weaknesses:**

Firstly, possibly due to time constraints, I feel that **this paper is not complete**. I read the article twice carefully. During the first read, I thought I had missed many details. However, upon a second, more thorough review, I found that the paper contains **only fragmented descriptions of the training process** and **lacks any explanation of how inference is conducted**.
I believe the potential issues are as follows:

### 1. Unclear Architecture Representation
Figure 2 lacks clarity in its representation. Several concerns need to be addressed:
- The distinction between Position Part and Content Part is not well-defined
- In Figure 3, it's unclear why the Position Part from the Encoder is directly fed as input - is this representing the feature blocks from Image Features or fused features?
- The nature and purpose of the Content Part input need clarification
- There's an unexplained discrepancy between Figure 2 (5 outputs) and Figure 3 (3 outputs)

### 2. Attention Mechanism Implementation
Section 3.3 presents several implementation issues:
- The matrix $I$ is described as a filter for large detection queries in self-attention, but the implementation details are unclear
- The relationship between $O_{txt}$ (final output) and its application in previous self-attention layers needs elaboration
- The specific self-attention layer where the filtering is applied is not specified
- The Content Isolation mechanism needs clarification, particularly regarding:
  - Its operation between detection and track queries
  - The implementation details
  - Whether it uses category features for query filtering

### 3. Dual-Branch Decoder Components
In Section 3.4:
- The origins of matrices $P$ and $O$ are not shown in the Dual-Branch Decoder diagram
- The purpose and utilization of $C^{(t+1)}$ obtained through propagation are not explained in the paper

### 4. Sequential Loss Computation
Section 3.5 presents logical inconsistencies:
- The sudden introduction of $P$ in the Sequential loss computation lacks context
- The relationship between this $P$ and the position part query from Section 3.3 needs clarification
- The connection between $L$ in Equation 6 and $L$ in Equation 5 is ambiguous

### 5. Inference Process
The paper lacks a comprehensive explanation of:
- The complete inference pipeline
- The feature fusion methodology for matching
- The integration of multiple features during inference

### 6. Claims of Mutual Benefits
The authors claim that "Our approach jointly models localization, appearance, and classification, enabling tracking and classification to mutually benefit from each other." However, this mutual benefit is not convincingly demonstrated:
- On the Validation Set:
  - Only ClsA shows improvement for Novel categories
  - Other metrics merely maintain similar performance levels
  - LocA actually decreases by one point
- On the Test Set:
  - Novel category classification improvement is negligible
  - Shows significant asymmetry compared to Validation Set results
  - The claimed mutual benefits are neither stable nor consistently demonstrated

### 7. Weak Experimental Results
The experimental results raise several concerns:
- Minimal improvement of only ~1 point in TETA for Novel categories
- Base TETA results are lower than OVTrack by one point on Validation Set
- The authors' claim of using only Base proposals is questionable given:
  - No improvement in Base category performance
  - Previous methods using unlabeled proposals achieve superior results without requiring additional annotations
  - The only apparent advantage seems to be computational efficiency

### 8. Methodology Clarification Needed
- The specific methodology for generating image sequences through data augmentation requires explanation

### 9. Inconsistency in Training Description
- Section 4.3 states "All methods are trained are trained solely on novel categories"
- This appears to be a typo and should likely read "solely on base categories"
- This inconsistency needs clarification

### 10. Ablation Study Concerns
The ablation experiments only using 10,000 images show:
- Generally low performance metrics compared with the results in Table 1
- Particularly low ClsA for base categories
- Limited discriminative power across different ablation settings

### 11. Visualization Issues
The supplementary materials' visualizations are problematic:
- The pole case example lacks representativeness
- The comparison with OVTrack is unclear (the right-side detection in the first row appears superior)
- The elephant case visualization is ambiguous
- No guidance is provided to help readers interpret the differences
- The dense bounding box visualizations make quality comparison difficult

**Questions:**

I have included all the questions in the 'Weaknesses section', outlining approximately twelve areas of questions.

---

> ### Author Response · Authors · 2024-11-27
> **Response to Reviewer 3Y3k (Part 1)**
>
> We appreciate your thoughtful effort in reviewing our work and for pointing out several detailed issues. We acknowledge that certain aspects of our writing may have caused confusion and have since revised the paper. The updated rebuttal revision presents all modifications in  **red font** .
>
> Thank you for raising these detailed questions, and we will address each of them comprehensively below.
>
> - **Q0. The paper contains only fragmented descriptions of the training process and lacks any explanation of how inference is conducted.**
>
>   Thank you for your thorough review, careful reading of our paper, and feedback on the clarity of the training and inference descriptions. We first provide a systematic and as easily understandable as possible description of the entire training process here, along with additional details about the inference process. Additionally, we have revised the methods section in the rebuttal revision to include a more structured explanation, clearly distinguishing between training and inference. These revisions are marked with **red text** and diagrams annotated with **red numbering**. We hope this improves the clarity of the paper and effectively communicates our innovative approach.
>
>   **Training process：** Our model is illustrated in **Figure 2 of the updated manuscript** (with added details). **We follow the general structure and basic tracking mechanism of MOTR, treating MOT as an iterative sequence prediction problem.** **Each trajectory is represented by a track query**. (**The basic structure and mechanisms related to MOTR are provided in Sec.3.1 of the rebuttal revision, beginning from line 158**)
>
>   Let us first revisit the representative iterative query-based tracker, MOTR. Following a DETR-like structure, detect queries  $Q_\text{det}^{t=1}$  for the first frame $f_{t=1}$ are fed into the Transformer decoder, where they interact with the image features $E_\text{img}^{t=1}$ extracted by the Transformer encoder. This process yields updated detect queries $Q_\text{det}^{\prime\,{t=1}}$ that contain object information. Detection predictions, including bounding boxes $B_\text{det}^{t=1}$ and object representations $O_\text{det}^{t=1}$, are subsequently extracted from $Q_\text{det}^{\prime\,{t=1}}$.
>
>   In contrast to DETR, for the query-based iterative tracker, $Q_\text{det}^{t=1}$ are only needed to detect newly appeared objects in the current frame. Consequently, one-to-one assignment is performed through bipartite matching exclusively between $Q_\text{det}^{\prime\,{t=1}}$ and the ground truth of the newly appeared objects, rather than matching with the ground truth of all objects.
>
>   The matched $Q_\text{det}^{\prime\,{t=1}}$ will be used to update and generate the track queries $Q_\text{tr}^{t=2}$, which, for the second frame $f_{t=2}$, are fed once again into the Transformer decoder and interact with the image features $E_\text{img}^{t=2}$ to obtain the representations and locations of the objects matched with $Q_\text{tr}^{ t=2}$, thereby enabling tracking predictions. The $Q_\text{tr}^{t=2}$ maintain their object associations and are updated to generate the $Q_\text{tr}^{t=3}$ for the third frame $f_{t=3}$. Parallel to $Q_\text{tr}^{t=2}$, and similar to the process for $f_{t=1}$, $Q_\text{det}^{t=2}$ are fed into the decoder to detect newly appeared objects. $Q_\text{det}^{\prime\,{t=2}}$ undergo binary matching, and the matched queries are transformed into new track queries, which are then added to $Q_\text{tr}^{t=3}$ for $f_{t=3}$. The entire tracking process can be extended to subsequent frames.
>
>   Regarding optimization, MOTR employs multi-frame optimization, where the loss is computed by considering both ground truths and matching outcomes. The matching results for each frame include both the maintained track associations and the binary matching results between $Q_\text{det}^\prime $ and newly appeared objects.
>
>   Leveraging the iterative nature of the query-based framework, OVTR transfers information about tracked objects across frames, aggregating category information throughout continuous image sequences to achieve robust classification performance, rather than performing independent localization and classification in each frame.

---

> ### Author Response · Authors · 2024-11-27
> **Response to Reviewer 3Y3k (Part 2)**
>
> (Continued from Part 1)
>
>
>   In the encoder, preliminary image features from the backbone and text embeddings from the CLIP model are processed through pre-fusion to generate fused image features $E_{img}$ and text features $E_{txt}$. We propose a dual-branch decoder comprising the OFA branch and the CTI branch. Upon input of $Q = [Q_\text{det}, Q_\text{tr}]$, the two branches respectively guide $Q$ to derive visual generalization representations and perform deep modality interaction with $E_\text{txt}$, outputting $O_\text{img}$, $O_\text{txt}$. $O_\text{img}$ serve as the input for the category information propagation (CIP) strategy, injecting category information into the category information flow. This process is an extension of the aforementioned mechanism where $Q_\text{det}^{\prime\,t}$ generates $Q_\text{tr}^{t+1}$. Meanwhile, $O_\text{txt}$ are utilized for computing category logits and for contrastive learning.
>
>   **The details in sections 3.2, 3.3, 3.4, and 3.5 have been supplemented to provide a clearer and more understandable explanation of the method. Changes are visible in red font.**
>
>   **Inference Process:** During inference, The network forward process during inference follows the same procedure as during training, with both detect queries $Q_\text{det}$ and track queries $Q_\text{tr}$ input in parallel. The key difference lies in the conversion of tracking queries. In detection predictions, if the category confidence score exceeds $\tau_\text{det}$, the corresponding updated detection query is transformed into a new tracking query, effectively initiating a new track. Conversely, if a tracked object is lost in the current frame (confidence $\leq \tau_\text{tr}$), it is marked as an inactive track. If an inactive track is lost for $T_\text{miss}$ consecutive frames, it is completely removed.
>
>   We hope that within the limited space here, we have provided as clear an explanation of our work as possible. More details can be found in the updated method section, marked in red, and in the supplementary materials (**Appendix A C D** of the rebuttal revision).
> - **Q1. Unclear Architecture Representation. Figure 2 lacks clarity in its representation. Several concerns need to be addressed:**
>
>   > Q1.1 The distinction between Position Part and Content Part is not well-defined
>   >
>   > Q1.2 In Figure 3, it's unclear why the Position Part from the Encoder is directly fed as input - is this representing the feature blocks from Image Features or fused features?
>   >
>   > Q1.3 The nature and purpose of the Content Part input need clarification
>   >
>   > Q1.4 There's an unexplained discrepancy between Figure 2 (5 outputs) and Figure 3 (3 outputs)
>   >
>
>   We have revised **Figure 2** and **Figure 3** in the manuscript based on your feedback. These figures have been adjusted to provide a more symbolic representation, clearly indicating the relationships between the different variables, as well as the inputs and outputs.
>
>   We would first like to clarify that the detection part of our model is based on DETR, and for the extraction and initialization of queries, we adopted the approach from DINO-DETR. This is mentioned in Section 3.2 of the **Rebuttal Revision**, with the relevant part highlighted in red font.
>
>   [**Q1.1** ] In Figure 2, we added the boundary isolation between the Position Part and Content Part of the queries, as well as the relationship between the Position Part and the encoder. The relationship of queries being jointly input into the decoder is also clarified in the annotation on the right side of the figure.
>
>   [**Q1.2** ] We have revised Figure 3 to make the expression of the Position Part feed clearer. Specifically, we adopted the sin-cos positional encoding approach from DINO-DETR. The positions with the top K classification scores extracted from the encoder’s output $E_\text{img}$, are selected as reference points. These are then processed using sin-cos positional encoding to generate position embeddings, which represent the Position Part of the queries. This enables the Position Part to incorporate preliminary object location information, which is common in DETR-based models such as DINO-DETR and two-stage Deformable DETR.
>
>   [**Q1.4** ] In Figure 3, we clarified the positions of outputs not reflected in Figure 2. $O_\text{img}$ represents the output of the OFA branch's feed-forward network, while $P^{\prime}$ (we now use $P^{\prime}$ to distinguish the updated Position Part from the previous one $ P$) is generated by the box head and sin-cos positional encoding from $O_\text{img}$.
>
>   Additionally, [**Q1.1** ] [ **Q1.3(purpose)** ] In DETR-based detectors or trackers, the position part and content part of the queries typically serve distinct roles: the position part is responsible for retrieving the locations of the objects, while **the content part focuses on and extracts the features of the objects** from the image.

---

> ### Author Response · Authors · 2024-11-27
> **Response to Reviewer 3Y3k (Part 3)**
>
> (Continued from Part 2)
>
>   Therefore, [**Q1.3(nature)** ] for the content part of the queries, we chose to use the learnable initialization approach, similar to DINO-DETR. This is because the features extracted in the encoder are preliminary content features that have not been further refined, which could potentially introduce ambiguity and mislead the decoder, especially as they may contain semantic information that could lead to incorrect classifications.
>
>   *[1] Zhang, Hao, et al. "DINO: DETR with Improved DeNoising Anchor Boxes for End-to-End Object Detection." The Eleventh International Conference on Learning Representations.
>   [2] Zhu, Xizhou, et al. "Deformable DETR: Deformable Transformers for End-to-End Object Detection." International Conference on Learning Representations.*
> - **Q2. Attention Mechanism Implementation**
>
>   > Q2.1 The matrix $I$ is described as a filter for large detection queries in self-attention, but the implementation details are unclear
>   >
>   > Q2.2 The relationship between $O_{txt}$ (final output) and its application in previous self-attention layers needs elaboration
>   >
>   > Q2.3 The specific self-attention layer where the filtering is applied is not specified
>   >
>   > Q2.4 The Content Isolation mechanism needs clarification, particularly regarding:
>   >
>   > &emsp;Q2.4.1 Its operation between detection and track queries
>   >
>   > &emsp;Q2.4.2 The implementation details
>   >
>   > &emsp;Q2.4.3 Whether it uses category features for query filtering
>   >
>
>   In response to your question, we provide a detailed explanation here. Additionally, a specific introduction to the two isolation strategies can be found **in Section A.4, 'Details of Attention Isolation Strategies,' of the Appendix, beginning from line 692**. Strictly speaking, we prefer to refer to them as masks or filters for attention weights, rather than filters for the queries, as we do not discard any queries during the self-attention phase.
>
>   [**Q2.1**] For the category isolation mask $I$, $I$ is **added to the attention weights** obtained from the batch matrix multiplication between Q and K during the self-attention process [**Q2.3**] in decoder layers 2 to 6 of the Transformer decoders. After this addition, a softmax operation is applied, and the result is multiplied by the value. This mechanism effectively masks attention between queries with significantly different category information, preventing mutual interference. Note: This mask is not used in the first decoder layer, as the content part of the queries fed into the self-attention in the first decoder layer is initialized through learnable parameters and does not contain category information. Detailed explanations are provided in the updated supplementary materials. **Detailed explanations are provided in Appendix A.4 of the rebuttal revision, beginning from line 694.**
>
>   [**Q2.2**]
>   - $O_{\text{txt}}$ is the output of the CTI (Category Text Interaction) branch. It is the direct representation obtained from the Aligned Queries and Text Features through cross-attention, followed by a feed-forward network (FFN), without any additional network heads.
>   - When $O_{\text{txt}}$ is output from the current decoder layer and passed into the next decoder layer (following the standard six-layer decoder setup in DETR-like models), $O_{\text{txt}}$ will serve as the content part of the queries fed into the self-attention of the next decoder layer (this is why the category isolation mask is necessary, as $O_{\text{txt}}$ already contains category information). For the next decoder layer, we can use a symbolic representation similar to the ones updated in our rebuttal revision. Just as the input queries for the first decoder layer are represented as $Q_{l1} = [P_{l1}, C_{l1}]$, for the second decoder layer we also have $Q_{l2} = [P_{l2}, C_{l2}]$, where $C_{l2}=O_{\text{txt},{l1}}$ and $P_{l2} =P^\prime_{l1}$(updated position part). The category isolation mask $I$, calculated based on the predicted categories from $O_{\text{txt},{l1}}$, is applied to the attention weights of the self-attention in the next decoder layer, as previously mentioned in Section 3.3 and now in **Line 292 of Section 3.3** in the rebuttal revision. This illustrates **the relationship between $O_{\text{txt}}$ and the self-attention layer** in the decoder.
>   For further details, please refer to **the first paragraph of Appendix A.4** in the rebuttal revision. We have also added an explanation in Appendix D regarding the standard setup of the six decoder and six encoder layers, as well as the transmission of queries between the decoders.

---

> ### Author Response · Authors · 2024-11-27
> **Response to Reviewer 3Y3k (Part 4)**
>
> (Continued from Part 3)
>   - Additionally, the self-attention in the decoder layer, based on the DETR-like structure, facilitates the interaction of information between queries and helps suppress duplicate detection boxes without requiring NMS. In our approach, self-attention enables queries to focus on the object information between each other, improving the global prediction of objects in the image. Queries with similar category information are more strongly interacted with (as attention is based on similarity), leading to better information exchange and enhanced global perception of the category. Therefore, the previous self-attention layer can capture more global category information during the process of generating $O_{\text{txt}}$, enhancing the stability of the category information in $O_{\text{txt}}$.
>
>
>   [**Q2.3**] As mentioned earlier, the category isolation mask is applied in the self-attention of decoder layers 2 to 6. The content isolation mask is used in the self-attention of the first decoder layer. A detailed explanation can be found in Appendix A.4 of the rebuttal revision, beginning from line 694.
>
>   [**Q2.4.1**] Our approach, as a MOTR-like tracker, concatenates the detect queries and track queries, which are then jointly fed as queries into the decoder for self-attention. This allows for information exchange between the queries in decoder layers 2 to 6, but the interaction between detect queries and track queries is blocked in the first decoder layer (as the first layer uses the content isolation mask in self-attention). In subsequent attention mechanisms, the queries participate in parallel computations without additional interactions. Specific details can be found in the updated Section 3.3 and Appendix A.4.
>
>   [**Q2.4.2**] The usage details of the content isolation mask are the same as those of the category isolation mask, with the only difference being the layers in which they are applied—specifically, the first decoder layer for the content isolation mask and layers 2 to 6 for the category isolation mask. For clarity, we have made adjustments in **Appendix A.4 of the rebuttal revision, beginning from line 721**
>
>   [**Q2.4.3**]The implementation of the content isolation mask is straightforward and does not involve category features.
>
> - **Q3. Dual-Branch Decoder Components**
>
>   > Q3.1 The origins of matrices $P$ and $O$ are not shown in the Dual-Branch Decoder diagram
>   >
>   > Q3.2 The purpose and utilization of $C^{t+1}$ obtained through propagation are not explained in the paper
>
>   [**Q3.1**]
>
>   - The generation of$P$ is explained in the response to [Q1.2] in A1. Specifically, we adopted the sin-cos positional encoding approach from DINO-DETR. The positions with the top K classification scores extracted from the encoder's output $E_\text{img}$, are selected as reference points. These are then processed using sin-cos positional encoding to generate position embeddings, which represent the Position Part $P$ of the queries. We have illustrated the position of $P$ in the updated Dual-Branch Decoder diagram.
>   - $O_\text{img}$ is the output of the feed-forward network in the OFA branch (we have updated Figure 3 and included it in the revised Rebuttal Revision).
>   - $O_{\text{txt}}$ is the output of the CTI (Category Text Interaction) branch. It is the direct representation obtained from the Aligned Queries and Text Features through cross-attention, followed by a feed-forward network (FFN), without any additional network heads.
>
>   [**Q3.2** ] Both $C^{t+1}$ and $C^{t}$ represent the same type of representation (with $C^{t}$ explained in Equation (4)), which is the content part of the queries. The difference is that $C^{t}$ is used when processing the current frame $f_{t}$, while $C^{t+1}$ is used when processing the next frame $f_{t+1}$. We have provided a clearer explanation in Section 3.4 of the rebuttal revision, **starting from line 322**.
>
>   Consider the relationship between track queries and detect queries, as well as the queries updated to track queries for the next frame, which are selected based on the matching results. We now use more accurate notation, $C_{*}^{t}$ and $C_\text{tr}^{t+1}$. The * represents the content part of the updated matched queries, $t$ indicates the frame number, and $tr$ refers to the content part of track queries. This change aligns with the updated method overview detailed in Section 3.1 of our rebuttal revision, **starting from line 158**.

---

> ### Author Response · Authors · 2024-11-27
> **Response to Reviewer 3Y3k (Part 5)**
>
> (Continued from Part 4)
>
> - **Q4. Sequential Loss Computation**
>
>   > Q4.1 The sudden introduction of $P$ in the Sequential loss computation lacks context
>   >
>   > Q4.2 The relationship between this $P$ and the position part query from Section 3.3 needs clarification
>   >
>   > Q4.3 The connection between $L$ in Equation 6 and $L$ in Equation 5 is ambiguous
>
>
>   [**Q4.1** ] [ **Q4.2** ] Apologies for the repeated use of the symbol; thank you for pointing this out. In Section 3.5, $P$ refers to the ground truth, and we have updated the symbol $P$ to $y$ in the revised Rebuttal Revision. The changes are highlighted in red.
>
>   [**Q4.3** ] In fact, $\mathcal{L}$ in Equation (6) and $\mathcal{L}$ in Equation (5) both represent the total loss for a single frame. $\mathcal{L}_{seq}$ represents the loss computation for a multi-frame image sequence, composed of the total losses of individual frames. The detailed expression of $\mathcal{L}$ in Equation (5) is provided to clarify that $\mathcal{L}$ is computed between the predictions and ground truth.
> - **Q5. Inference Process**
>
>   > Q5.1 The complete inference pipeline
>   >
>   > Q5.2 The feature fusion methodology for matching
>   >
>   > Q5.3 The integration of multiple features during inference
>
>   [**Q5.1** ] We provide an explanation of the **inference process** in response to  **Q0**. We have updated this in Section 3.1 of the rebuttal revision, specifically from lines 193 to 198.
>
>   [**Q5.2** ] In our method, feature fusion and matching are not directly related.
>
>   Feature fusion occurs in two places within our framework. First, in the encoder's fusion layer, it performs an initial fusion of image and text features to strengthen the representations of both modalities and help align features from different modalities. Second, in the dual-branch decoder, the Text Cross-Attention layer enables deep interaction between the aligned queries with visual generalization capabilities and the enhanced text features. This allows the aligned queries to focus on the category information in the text features, improving classification performance.
>
>   Matching refers to the training stage of OVTR, where detection queries are only used to detect newly appeared objects in the current frame. A one-to-one binary matching is applied between the updated detection queries $Q_\text{det}^{\prime}$ **and the ground truth of the newly appeared objects (rather than all objects). The matching results are then used to compute the loss with respect to the ground truths. Subsequently, the matched $Q_\text{det}^{\prime}$** are transformed into track queries for the next frame.
>
>   We consider that that it might also be the calculation of the alignment loss in Section 3.2, Dual-Branch Structure, where it is mentioned that '$F_\text{align}$ corresponds to the results of the bipartite matching,' that caused your confusion. To clarify this point, we have added a reference (rebuttal revision line 262 in red) directing to **Section 3.1 Overview (lines 179 to 192)**, where we further elaborate on the role of matching. We hope this addition addresses your concerns.
>
>   [**Q5.3**] As explained in the **Inference Process** section of  **Q0** , the inference process follows the same network forward process as the training process. Both share the same multimodal fusion and category information flow aggregation methods.
>
> - **Q6. Claims of Mutual Benefits**
>
>   > On the Validation Set:
>   >
>   > &emsp;Only ClsA shows improvement for Novel categories
>   >
>   > &emsp;Other metrics merely maintain similar performance levels
>   >
>   > &emsp;LocA actually decreases by one point
>   >
>   > On the Test Set:
>   >
>   > &emsp;Novel category classification improvement is negligible
>   >
>   > &emsp;Shows significant asymmetry compared to Validation Set results
>   >
>   > &emsp;The claimed mutual benefits are neither stable nor consistently demonstrated
>   >
>
>   - First,**our model is not an improvement based on OVTrack; OVTrack is not our baseline.** Comparing it with OVTrack does not demonstrate whether our model allows classification and tracking to benefit from each other.
>   - We adjusted the model's training strategy,  **leading to enhanced performance**. Specifically, the number of training epochs was increased from 12 to **15**. At the 13th epoch, the learning rate underwent an additional decay by a factor of 10. Multi-frame training was adjusted, increasing the number of frames from 2 to 3, 4, and **5** at the 4th, 7th, and **14th** epochs, respectively (previously 2, 3, and 4 at the 4th and 7th epochs). This allowed the model to accurately locate and track objects over longer video sequences. The performance of the model trained with this strategy is presented in the table below.

---

> ### Author Response · Authors · 2024-11-27
> **Response to Reviewer 3Y3k (Part 6)**
>
> (Continued from Part 5)
>
> The results demonstrate that, compared to OVTrack, our method achieved a **12.9%** improvement on the TETA metric for novel categories and a **3.1%** improvement for base categories on the validation set. On the test set, TETA increased by **12.4%** for novel categories and **5.8%** for base categories.
>
> | **Validation set** | Data         | ${\text{Proposals}}_{\text{novel}}$ | TETA↑(novel)  | LocA↑(novel)  | AssocA↑(novel) | ClsA↑(novel) | TETA↑(base)   | LocA↑(base)   | AssocA↑(base) | ClsA↑(base)   |
> | -- | -- | --- | -- | -- | --- | --- | -- | -- | -- | -- |
> | OVTrack                  | G-LVIS, LVIS | ✓                                    | 27.8           | 48.8           | 33.6            | 1.5           | 35.5           | 49.3           | 36.9           | **20.2** |
> | OVTR                     | LVIS         |                                       | **31.4** | **54.4** | **34.5**  | **5.4** | **36.6** | **52.2** | **37.6** | 20.1           |
>
> | **Test set** | Data         | ${\text{Proposals}}_{\text{novel}}$ | TETA↑(novel)  | LocA↑(novel)  | AssocA↑(novel) | ClsA↑(novel) | TETA↑(base)   | LocA↑(base)   | AssocA↑(base) | ClsA↑(base)   |
> | -- | -- | --- | -- | -- | --- | --- | -- | -- | -- | -- |
> | OVTrack            | G-LVIS, LVIS | ✓                                    | 24.1           | 41.8           | 28.7            | 1.8           | 32.6           | 45.6           | 35.4           | **16.9** |
> | OVTR               | LVIS         |                                       | **27.1** | **47.1** | **32.1**  | **2.1** | **34.5** | **51.1** | **37.5** | 14.9           |
>   - Additionally, regarding the symmetry between the validation set and the test set, **OVTrack is not our baseline, nor is our model an improvement upon it**. Therefore,**it is expected that our performance testing exhibits a different performance distribution**.
>   - To address your **concerns about the mutual benefit of classification and tracking** in our model, we have conducted ablation experiments to demonstrate this.
>     Ablation study on OVTR components:
>
>     | CIP | Isolation | Dual | TETA           | AssocA         | ClsAb          | ClsAn         |
>     | --- | --- | -- | -- | -- | -- | --- |
>     |     |           |      | 28.9           | 29.1           | 10.7           | 1.7           |
>     | √  |           |      | 30.3           | 30.2           | 12.1           | 1.8           |
>     | √  | √        |      | 31.2           | 31.8           | 12.6           | 1.9           |
>     | √  |           | √   | 32.1           | 32.8           | 14.6           | 2.3           |
>     | √  | √        | √   | **33.2** | **34.5** | **15.2** | **2.7** |
>
>     **The first row** represents the baseline model we constructed, which **does not incorporate category information when iterating track queries—essentially, it does not establish a category information flow**. Specifically, in the single-branch structure (CTI only), we added an auxiliary network structure identical to the standard MOTR decoder, but it operates entirely independently of the CTI branch. As a result, the queries passed through this structure lack category information. When using its output as input to CIP, the category information flow lacks direct category information. **Dual** indicates the dual-branch structure, and **Isolation** represents the protective strategies within the decoder.
>
>     As seen in the results, when category information is not used to update the Track queries—i.e., when no direct category information is input into the CIP (first row)—the association and classification scores are significantly lower compared to when category information is input into the CIP (second row). This demonstrates that our tracking queries benefit from category information to achieve more stable tracking. Additionally, during multi-frame classification, the category prior information in the track queries enhances the classification performance.
>
> - **Q7. Weak Experimental Results**
>
>   > Q7.1 Minimal improvement of only ~1 point in TETA for Novel categories
>   >
>   > Q7.2 Base TETA results are lower than OVTrack by one point on Validation Set
>   >
>   > Q7.3 The authors' claim of using only Base proposals is questionable given:
>   >
>   > &emsp;Q7.3.1 No improvement in Base category performance
>   >
>   > &emsp;Q7.3.2 Previous methods using unlabeled proposals achieve superior results without requiring additional annotations
>   >
>   > &emsp;Q7.3.3 The only apparent advantage seems to be computational efficiency
>   >
>
>   [**Q7.1**] [**Q7.2**]The updated evaluation results can be found in  **Q6** , showing that our method achieves higher scores than OVTrack for both novel TETA and base TETA.

---

> ### Author Response · Authors · 2024-11-27
> **Response to Reviewer 3Y3k (Part 7)**
>
> (Continued from Part 6)
>
>
>   [**Q7.3.1**]**OVTrack is not our baseline, nor is our model an improvement based on it. The difference in score distribution between our novel and base results and those of OVTrack is expected.** We attribute the higher novel TETA metric primarily to the superior generalization of our model. While earlier versions showed relatively weaker performance on the base metric, our current results surpass OVTrack in both the base and novel TETA metrics.
>
>   [**Q7.3.2**]
>
>   * In fact, **we did not use any additional annotations**. Instead, we strictly adhered to the original annotations of the dataset to generate the image embeddings, **following a more rigorous interpretation of the open-vocabulary definition**.
>   * In contrast, OVTrack utilized an additional RPN-based detector to pre-extract a large number of proposals (99.4M) before training. These proposals were then converted into embeddings through a CLIP image encoder (over 100 GB in size) and used as distillation targets during training (including novel categories). The specific quantity can be found in the open-source project, and the method is detailed in the original ViLD paper.
>
>   *[3] Gu, Xiuye, et al. "Open-vocabulary Object Detection via Vision and Language Knowledge Distillation." International Conference on Learning Representations.*
>
>   [**Q7.3.3**] Our current version achieves both high computational efficiency and strong performance. The computational efficiency is validated as follows:
>
>   We evaluated the inference speed of OVTrack and OVTR on a single NVIDIA GeForce RTX 3090 GPU. The results, summarized in the table below, indicate that OVTR achieves faster inference compared to OVTrack. Additionally, we tested a lightweight version, OVTR-Lite, which excludes the category isolation strategy and tensor KL divergence computation. Despite some performance trade-offs, OVTR-Lite still outperforms OVTrack in overall performance. It achieves nearly **4× faster inference speed** while maintaining GPU memory usage within **4GB** during inference.
>
>   | **Validation set** | **FPS**  | TETA↑(novel)  | LocA↑(novel)  | AssocA↑(novel) | ClsA↑(novel) | TETA↑(base)   | LocA↑(base)   | AssocA↑(base) | ClsA↑(base)   |
>   | -- | -- | -- | -- | --- | --- | -- | -- | -- | -- |
>   | OVTrack                  | 3.1            | 27.8           | 48.8           | 33.6            | 1.5           | 35.5           | 49.3           | 36.9           | **20.2** |
>   | OVTR                     | **3.4**  | **31.4** | **54.4** | **34.5**  | **5.4** | **36.6** | **52.2** | **37.6** | 20.1           |
>   | **OVTR-Lite**      | **12.4** | 30.1           | 52.7           | 34.4            | 3.14          | 35.6           | 51.3           | 37.0           | 18.6           |
>
>   In summary, despite being transformer-based, OVTR achieves relatively fast inference speed due to its end-to-end nature, which eliminates the need for complex post-processing and explicit similarity associations between frames.
>
>   We want to emphasize that we propose a novel open-vocabulary framework, aiming to bring **a new perspective** of end-to-end and modality interaction fusion **to the OVMOT field**. At the same time, we believe that many new methods and ideas could enhance our work, giving our approach **significant potential** for improvement. Furthermore, we are confident that there is **considerable room for performance enhancement**, as demonstrated by the improvements brought by our adjusted training strategy.
>
>   Not only does our method excel in performance, but its advantages can be summarized as follows:
>
>
>   - **1**. Our method is end-to-end, **eliminating the need for complex post-processing and explicit similarity associations between frames**. Additionally, our model achieves faster inference speeds compared to OVTrack, as detailed in our response to [ **Q7.3.3** ].
>   - **2**. OVTrack requires off-the-shelf OVDs and necessitates additional tuning and training of two models. This becomes particularly challenging in diverse open-vocabulary scenarios, where tuning can be difficult.
>   - **3**. Our method **does not require a separate RPN-based detector to pre-extract proposals** (which may contain novel class information, but our method does not rely on it). Additionally, it does not require using the CLIP image encoder to generate embeddings for proposals, which can result in hundreds of gigabytes of embeddings. This significantly saves time and reduces operations during the preprocessing stage.
>   - **4**. In implementing open-vocabulary tasks, our method does not rely on the presence of a large number of unannotated novel category objects in the training dataset, reducing the requirements on the dataset.
>   - **5**. Our method does not rely on the diffusion model mentioned in OVTrack to double the data size, thus reducing the need for large training datasets.

---

> ### Author Response · Authors · 2024-11-27
> **Response to Reviewer 3Y3k (Part 8)**
>
> (Continued from Part 7)
>
> - **Q8. Methodology Clarification Needed**
>
>   > The specific methodology for generating image sequences through data augmentation requires explanation
>   >
>
>   Our data augmentation includes conventional techniques, such as applying random resizing, horizontal flipping, color jittering, and random affine transformations to single images to create distinguishable multi-frame data. This part aligns with OVTrack. Additionally, we propose **Dynamic Mosaic** and **Random Occlusion** augmentations, specifically designed for MOTR-like trackers. Detailed explanations, illustrations, and examples of these methods have been added to the supplementary materials in the updated rebuttal revision.
>
>   Specifically, while query-based trackers excel at maintaining associations over extended periods, they place higher demands on the realism of object motion patterns in training data. For OVTR, track queries must not only learn to capture the same object as it moves to a new position in the next frame, but also handle scenarios such as**object appearance** and  **disappearance** ,  **tracking continuity after occlusion** , and maintaining correct associations when  **relative motion occurs between tracked objects and others** .
>
>   To address these challenges, we propose**Dynamic Mosaic** augmentation, an improvement over the Mosaic augmentation in OVTrack. In addition to stitching four different images into a single composite, Dynamic Mosaic generates images with varying relative spatial relationships among objects across different training frames. This simulates scenarios such as objects approaching or receding from each other, crossing paths, and exhibiting relative size changes. The **Random Occlusion** augmentation is employed to simulate situations where objects disappear due to occlusion and then reappear or suddenly emerge in the scene.
>
>   More detailed augmentation techniques and illustrative examples can be found in **Appendix C** of the rebuttal revision, **starting from line 805**.
>
>   *[4] Li, Siyuan, et al. "Ovtrack: Open-vocabulary multiple object tracking." Proceedings of the IEEE/CVF conference on computer vision and pattern recognition. 2023.*
> - **Q9. Inconsistency in Training Description**
>
>   > Section 4.3 states "All methods are trained are trained solely on novel categories". This appears to be a typo and should likely read "solely on base categories" .This inconsistency needs clarification
>   >
>
>   Apologies for the typo in this section, and thank you for pointing it out. We have corrected it in the latest Rebuttal Revision.
>
> - **Q10. Ablation Study Concerns**
>
>   > Generally low performance metrics compared with the results in Table 1
>   >
>   > Particularly low ClsA for base categories
>   >
>   > Limited discriminative power across different ablation settings
>   >
>
>   We have **expanded the training data** for the ablation experiment by **4 times**, and the updated results are as follows.
>
>   - Table 3: Ablation study on decoder components：
>     | CTI | OFA | Align | TETA           | AssocA         | ClsAb          | ClsAn         |
>     | --- | --- | --- | -- | -- | -- | --- |
>     | √  |     |       | 31.2           | 31.8           | 12.6           | 1.9           |
>     | √  | √  |       | 31.9           | 32.0           | 13.9           | 2.1           |
>     | √  |     | √    | 32.5           | 33.9           | 13.8           | 2.0           |
>     | √  | √  | √    | **33.2** | **34.5** | **15.2** | **2.7** |
>   - Table 4: Ablation study on the protection strategies for the decoders：
>     | Category  | Content | TETA           | AssocA         | ClsAb          | ClsAn         |
>     | -- | --- | -- | -- | -- | --- |
>     |          |         | 32.1           | 32.8           | 14.6           | 2.3           |
>     | √       |         | 32.2           | 33.0           | **15.6** | 2.5           |
>     |          | √      | 32.4           | 33.6           | 14.3           | 2.5           |
>     | √       | √      | **33.2** | **34.5** | 15.2           | **2.7** |
>   - Table 5: Ablation study on alignment methods:
>     | Alignment    | TETA           | AssocA         | ClsAb          | ClsAn         |
>     | -- | -- | -- | -- | --- |
>     | Text         | 31.6           | 32.6           | 14.0           | 2.0           |
>     | Image (used) | **33.2** | **34.5** | **15.2** | 2.7           |
>     | Avg          | 32.3           | 33.2           | 14.1           | **3.2** |
>   - Table 6: Ablation study on inputs for CIP $I_\text{CIP}$:
>     | $I_\text{CIP}$ | TETA           | AssocA         | ClsAb          | ClsAn         |
>     | -- | -- | -- | -- | --- |
>     | Otxt               | 32.5           | 33.8           | 14.7           | 1.9           |
>     | Oimg(used)         | **33.2** | **34.5** | **15.2** | **2.7** |

---

> ### Author Response · Authors · 2024-11-27
> **Response to Reviewer 3Y3k (Part 9)**
>
> (Continued from Part 8)
>
>   - Table 7: Different Image Sequence Lengths for Multi-Frame Optimization:
>     | Length  | TETA           | AssocA         | ClsAb          | ClsAn         |
>     | --- | -- | -- | -- | --- |
>     | 2       | 27.9           | 24.9           | 13.5           | 2.2           |
>     | 3       | 30.5           | 32.2           | 14.5           | 2.5           |
>     | 4       | 31.6           | 33.8           | 14.7           | 2.4           |
>     | 5(used) | **33.2** | **34.5** | **15.2** | **2.7** |
>
>   From the results above, we observe that the ablation experiment conducted with 40,000 images yields relative performance among different ablation settings that is largely consistent with the results obtained from training on 10,000 images. This new result aligns with our previous analysis. **With the increased data volume, the overall evaluation results of the ablation experiments, particularly classification performance, have improved. The performance differences between models under different ablation settings have become more pronounced**, highlighting the advantages of these configurations. It can still be concluded that our proposed CIP, dual-branch structure, and decoder protection strategy contribute to significant performance improvements.
>
>   Additionally, we have conducted an ablation study without some components. The results can be found in **Q6** , which includes the experiments demonstrating mutual benefit.
>
>   The ablation study on OVTR components in **Q6** show that each component effectively enhances tracking performance. OVTR achieves a 14.9% improvement on TETA compared to the constructed baseline.The adoption of the CIP strategy increases AssocA by 3.8\% and $\text{ClsA}_\text{b}$ by 13.1\%, significantly improving both association and classification. This demonstrates that our model enables harmonious collaboration between tracking and classification. Additionally, the dual-branch structure and isolation strategies further enhance the model's open-vocabulary tracking performance, aligning with the findings from our earlier ablation experiments.
>
>   We have updated the analysis of the **Ablation Study on OVTR Components** in  **Appendix B.1** .
> - **Q11. Visualization Issues**
>
>   > Q11.1 The pole case example lacks representativeness
>   >
>   > Q11.2 The comparison with OVTrack is unclear (the right-side detection in the first row appears superior)
>   >
>   > Q11.3 The elephant case visualization is ambiguous
>   >
>   > Q11.4 No guidance is provided to help readers interpret the differences
>   >
>   > Q11.5 The dense bounding box visualizations make quality comparison difficult
>   >
>
>   [**Q11.1**] We have added more representative tracking visualizations of objects in the supplementary materials of the Rebuttal Revision. **(Appendix E, starting from line 1026)**
>
>   [**Q11.2**] We selected this set of images to demonstrate that our tracking achieves better association performance. The output results after updating the model training strategy to achieve higher LocA are included in the updated supplementary materials. (Appendix E, Figure 9, Figure 10)
>
>   [**Q11.3**] The text labels for the elephant outputs are smaller because the evaluation scenarios for open-vocabulary tracking typically involve numerous and complex objects. Using larger labels would obscure the objects in the image. Of course, we have also made sure to select more intuitive results.
>
>   [**Q11.4**] We have added an analysis of the tracking visualization results in the supplementary materials of the Rebuttal Revision. (Appendix E, starting from line 917)
>
>   [**Q11.5**] As mentioned in the response to [Q11.3], the evaluation **scenarios for open-vocabulary tracking generally involve numerous and complex objects**. Additionally, the open-vocabulary multi-object tracking task is specifically designed to apply to real-world scenarios, which are typically complex and involve many objects. We aim to showcase results that **better simulate real-world open-world situations**.
>
> We would like to express our sincere gratitude for the reviewer's hard work and the detailed questions raised. We have provided explanations, made revisions, and conducted additional experiments for each of the reviewer's comments.

---

> ### Author Response · Authors · 2024-12-03
> **Looking forward to the reviewer's feedback on the rebuttal**
>
> Dear Reviewer 3Y3k,
>
> We would like to express our sincere gratitude for your thoughtful and thorough review of our work. We have carefully addressed all of the questions you raised across the 12 areas, providing detailed responses that cover both the specific aspects of our method and the experimental results. Some of your questions may have arisen due to our limited explanation of certain prior knowledge related to OVTR (e.g., the mechanism of MOTR), which may not be as widely familiar within the MOT field. To clarify our approach, we have included some preliminaries on OVTR in the overview section.
>
> Thank you once again for your time and substantial input in reviewing our work. As the discussion phase nears its deadline, we would greatly appreciate it if you could revisit our rebuttals and let us know if they adequately address your concerns or if further clarifications are needed.
>
> We look forward to your feedback and hope you have a pleasant day ahead!
>
> Sincerely,
>
> The Authors

---

### Official Review · Reviewer_tyWH · 2024-11-04

**Soundness:** 3
**Presentation:** 2
**Contribution:** 3
**Rating:** 6
**Confidence:** 5

**Summary:**

In this paper, the authors propose OVTR (End-to-End Open-Vocabulary Multiple Object Tracking with TRansformer), the first
end-to-end open-vocabulary tracker.

Specifically, to achieve stable classification and continuous tracking, they designed the CIP (Category Information Propagation) strategy, which establishes multiple high-level category information priors for subsequent frames.

They also introduce a dual-branch structure for generalization capability and deep multimodal interaction and incorporate protective strategies in the decoder to enhance performance.

**Strengths:**

+ The motivation for this work, the flow of category information, is reasonable.

+ This is the first end-to-end framework for open-vocabulary tracking.

+ Experimental results have verified the effectiveness of the proposed method, to some extent.

**Weaknesses:**

- The writing of the method part is not easy to follow.

- From the results, we can see that the performance of the proposed method is not very excellent, compared to the basic baseline method OVTrack, not to mention the most recent methods.

- The authors are suggested to present the advantages of the end-to-end Transformer methods compared to the tracking-by-OVD methods, based on the experimental, including the performance, speed, model size, etc.

- The authors claim that due to resource constraints, the ablation studies are conducted on a subset of 10,000 images. This makes the evaluation not very convincing. The overall framework can be implemented on the whole dataset. The ablation study without some components can be also conducted.

- The object category in the KITTI dataset is not various, which is not very appropriate for the OVTrack task. The comparison of OVTrack and the proposed method on KITTI using Car Category is not very convective, for the OVTrack problem.

**Questions:**

See the weakness.

---

> ### Author Response · Authors · 2024-11-27
> **Response to Reviewer tyWH (Part 1)**
>
> We appreciate your detailed and valuable review on our paper.
>
> - **Q1. The writing of the method part is not easy to follow.**
>
>   Apologies for any confusion caused by the method section of our model. Here, we provide an overall explanation of the approach and model. Additionally, we have made adjustments in the updated rebuttal revision, addressing areas where the original method section was unclear. The revised parts are highlighted in red for easy identification.
>
>   Our model is illustrated in **Figure 2 of the updated manuscript** (with added details). **We follow the general structure and basic tracking mechanism of MOTR, treating MOT as an iterative sequence prediction problem.** **Each trajectory is represented by a track query**. (**The basic structure and mechanisms related to MOTR are provided in Sec.3.1 of the rebuttal revision, beginning from line 158.**)
>
>   Let us first revisit the representative iterative query-based tracker, MOTR. Following a DETR-like structure, detect queries  $Q_\text{det}^{t=1}$  for the first frame $f_{t=1}$ are fed into the Transformer decoder, where they interact with the image features $E_\text{img}^{t=1}$ extracted by the Transformer encoder. This process yields updated detect queries $Q_\text{det}^{\prime\,{t=1}}$ that contain object information. Detection predictions, including bounding boxes $B_\text{det}^{t=1}$ and object representations $O_\text{det}^{t=1}$, are subsequently extracted from $Q_\text{det}^{\prime\,{t=1}}$.
>
>   In contrast to DETR, for the query-based iterative tracker, $Q_\text{det}^{t=1}$ are only needed to detect newly appeared objects in the current frame. Consequently, one-to-one assignment is performed through bipartite matching exclusively between $Q_\text{det}^{\prime\,{t=1}}$ and the ground truth of the newly appeared objects, rather than matching with the ground truth of all objects.
>
>   The matched $Q_\text{det}^{\prime\,{t=1}}$ will be used to update and generate the track queries $Q_\text{tr}^{t=2}$, which, for the second frame $f_{t=2}$, are fed once again into the Transformer decoder and interact with the image features $E_\text{img}^{t=2}$ to obtain the representations and locations of the objects matched with $Q_\text{tr}^{ t=2}$, thereby enabling tracking predictions. The $Q_\text{tr}^{t=2}$ maintain their object associations and are updated to generate the $Q_\text{tr}^{t=3}$ for the third frame $f_{t=3}$. Parallel to $Q_\text{tr}^{t=2}$, and similar to the process for $f_{t=1}$, $Q_\text{det}^{t=2}$ are fed into the decoder to detect newly appeared objects. $Q_\text{det}^{\prime\,{t=2}}$ undergo binary matching, and the matched queries are transformed into new track queries, which are then added to $Q_\text{tr}^{t=3}$ for $f_{t=3}$. The entire tracking process can be extended to subsequent frames.
>
>   Regarding optimization, MOTR employs multi-frame optimization, where the loss is computed by considering both ground truths and matching outcomes. The matching results for each frame include both the maintained track associations and the binary matching results between $Q_\text{det}^\prime $ and newly appeared objects.
>
>   During inference: Similar to MOTR, the network forward process during inference in OVTR follows the same procedure as during training. The key difference lies in the conversion of track queries. In detection predictions, if the category confidence score exceeds $\tau_\text{det}$, the corresponding updated detect query is transformed into a new track query, initiating a new track. Conversely, if a tracked object is lost in the current frame (confidence $\leq \tau_\text{tr}$), it is marked as an inactive track. If an inactive track is lost for $T_\text{miss}$ consecutive frames, it is completely removed.
>
>
>   Leveraging the iterative nature of the query-based framework, OVTR transfers information about tracked objects across frames, aggregating category information throughout continuous image sequences to achieve robust classification performance, rather than performing independent localization and classification in each frame.
>
>   In the encoder, preliminary image features from the backbone and text embeddings from the CLIP model are processed through pre-fusion to generate fused image features $E_{img}$ and text features $E_{txt}$. We propose a dual-branch decoder comprising the OFA branch and the CTI branch. Upon input of $Q = [Q_\text{det}, Q_\text{tr}]$, the two branches respectively guide $Q$ to derive visual generalization representations and perform deep modality interaction with $E_\text{txt}$, outputting $O_\text{img}$, $O_\text{txt}$. $O_\text{img}$ serve as the input for the category information propagation (CIP) strategy, injecting category information into the category information flow. This process is an extension of the aforementioned mechanism where $Q_\text{det}^{\prime\,t}$ generates $Q_\text{tr}^{t+1}$. Meanwhile, $O_\text{txt}$ are utilized for computing category logits and for contrastive learning.

---

> ### Author Response · Authors · 2024-11-27
> **Response to Reviewer tyWH (Part 2)**
>
> (Continued from Part 1)
>
>
>   **The details in sections 3.2, 3.3, 3.4, and 3.5 have been supplemented to provide a clearer and more understandable explanation of the method. Changes are visible in red font.**
>
>   We hope that within the limited space here, we have provided as clear an explanation of our work as possible. More details can be found in the updated method section, marked in red, and in the supplementary materials (**Appendix A C D** of the rebuttal revision).
>
>   *[1]Zeng, Fangao, et al. "Motr: End-to-end multiple-object tracking with transformer."  *European Conference on Computer Vision* . Cham: Springer Nature Switzerland, 2022.*
> - **Q2. The performance of the proposed method is not very excellent, compared to the basic baseline method OVTrack, not to mention the most recent methods.**
>
>   Thanks for your valuable comment. After submission, we further iterated and optimized our existing training strategy, **resulting in improved performance**. We adjusted the training strategy of the model, **resulting in improved performance**. Specifically, the number of training epochs was increased from 12 to **15**. At the 13th epoch, the learning rate underwent an additional decay by a factor of 10. Multi-frame training was adjusted, increasing the number of frames from 2 to 3, 4, and **5** at the 4th, 7th, and **14th** epochs, respectively (previously 2, 3, and 4 at the 4th and 7th epochs). This allowed the model to accurately locate and track objects over longer video sequences. The performance of the model trained with this strategy is presented in the table below.
>
>
>   The results demonstrate that, compared to OVTrack, our method achieved a **12.9%** improvement on the TETA metric for novel categories and a **3.1%** improvement for base categories on the validation set. On the test set, TETA increased by **12.4%** for novel categories and **5.8%** for base categories. For localization accuracy (LocA), the validation set showed an improvement of **11.5%** for novel categories and **5.9%** for base categories, while the test set showed an increase of **12.7%** and **12.1%** for novel and base categories, respectively.
>
>   | **Validation set** | Data         | ${\text{Proposals}}_{\text{novel}}$ | TETA↑(novel)  | LocA↑(novel)  | AssocA↑(novel) | ClsA↑(novel) | TETA↑(base)   | LocA↑(base)   | AssocA↑(base) | ClsA↑(base)   |
>   | -- | -- | --- | -- | -- | --- | --- | -- | -- | -- | -- |
>   | OVTrack                  | G-LVIS, LVIS | ✓                                    | 27.8           | 48.8           | 33.6            | 1.5           | 35.5           | 49.3           | 36.9           | **20.2** |
>   | OVTR                     | LVIS         |                                       | **31.4** | **54.4** | **34.5**  | **5.4** | **36.6** | **52.2** | **37.6** | 20.1           |
>
>   | **Test set** | Data         | ${\text{Proposals}}_{\text{novel}}$ | TETA↑(novel)  | LocA↑(novel)  | AssocA↑(novel) | ClsA↑(novel) | TETA↑(base)   | LocA↑(base)   | AssocA↑(base) | ClsA↑(base)   |
>   | -- | -- | --- | -- | -- | --- | --- | -- | -- | -- | -- |
>   | OVTrack            | G-LVIS, LVIS | ✓                                    | 24.1           | 41.8           | 28.7            | 1.8           | 32.6           | 45.6           | 35.4           | **16.9** |
>   | OVTR               | LVIS         |                                       | **27.1** | **47.1** | **32.1**  | **2.1** | **34.5** | **51.1** | **37.5** | 14.9           |
>
>   At last, we would like to emphasize that OVTR is a novel paradigm of open-vocabulary MOT that achieves **OV tracking in a fully end-to-end manner.** The motivation behind this work is to establish a novel tracking paradigm distinct from the mainstream OVD-based tracking architectures. This stems from the fact that, while existing OVD-based architectures demonstrate decent tracking performance, they also have several inherent limitations. These include the need for an additional OV-Detector, complex post-tracking processing, and a lack of scalability to larger datasets, which collectively hinder further improvements and real-world applications.
>
>   In contrast, our proposed OVTR exhibits several advantageous characteristics: **a single model, fully end-to-end processing, no post-processing, and a data-friendly nature that facilitates scalability.** Thus, although our current approach does not significantly outperform existing OVD solutions at this stage, it has already demonstrated immense potential and **offers the OV-MOT community a novel research direction to explore**.

---

> ### Author Response · Authors · 2024-11-27
> **Response to Reviewer tyWH (Part 3)**
>
> (Continued from Part 2)
>
> - **Q3. The authors are suggested to present the advantages of the end-to-end Transformer methods compared to the tracking-by-OVD methods, based on the experimental, including the performance, speed, model size, etc.**
>
>   Thanks for your valuable suggestions. In fact, we have already discussed the differences and advantages of our method compared to tracking-by-OVD approaches in the introduction. Here, we provide further elaboration and a summary.
>
>   **1.** Our method is end-to-end, **eliminating the need for complex post-processing and explicit similarity associations between frames**. Additionally, our model achieves faster inference speeds compared to OVTrack.
>
>   We evaluated the inference speed of OVTrack and OVTR on a single NVIDIA GeForce RTX 3090 GPU. The results, summarized in the table below, indicate that OVTR achieves faster inference compared to OVTrack. Additionally, we tested a lightweight version, OVTR-Lite, which excludes the category isolation strategy and tensor KL divergence computation. Despite some performance trade-offs, OVTR-Lite still outperforms OVTrack in overall performance. It achieves nearly **4× faster inference speed** while maintaining GPU memory usage within **4GB** during inference.
>
>   | **Methods**   | **FPS**  | TETA↑(novel)  | LocA↑(novel)  | AssocA↑(novel) | ClsA↑(novel) | TETA↑(base)   | LocA↑(base)   | AssocA↑(base) | ClsA↑(base)   |
>   | --- | -- | -- | -- | --- | --- | -- | -- | -- | -- |
>   | OVTrack             | 3.1            | 27.8           | 48.8           | 33.6            | 1.5           | 35.5           | 49.3           | 36.9           | **20.2** |
>   | OVTR                | **3.4**  | **31.4** | **54.4** | **34.5**  | **5.4** | **36.6** | **52.2** | **37.6** | 20.1           |
>   | **OVTR-Lite** | **12.4** | 30.1           | 52.7           | 34.4            | 3.14          | 35.6           | 51.3           | 37.0           | 18.6           |
>
>   The speed test experiments demonstrating the inference speed of our model have been added in Appendix B.3 of our rebuttal revision, beginning from line 776.
>
>   **2.** OVTrack requires off-the-shelf OVDs and necessitates additional tuning and training of two models. This becomes particularly challenging in diverse open-vocabulary scenarios, where tuning can be difficult.
>
>   **3.** Our method **does not require a separate RPN-based detector to pre-extract proposals** (which may contain novel class information, but our method does not rely on it). Additionally, it does not require using the CLIP image encoder to generate embeddings for proposals, which can result in hundreds of gigabytes of embeddings. This significantly saves time and reduces operations during the preprocessing stage.
>
>   **4.** In implementing open-vocabulary tasks, our method does not rely on the presence of a large number of unannotated novel category objects in the training dataset, reducing the requirements on the dataset.
>
>   **5.** Our method does not rely on the diffusion model mentioned in OVTrack to double the data size, thus reducing the need for large training datasets.
>
>   In summary, we propose a novel open-vocabulary framework, aiming to bring **a new perspective** of end-to-end and modality interaction fusion to the OVMOT field. At the same time, we believe that many new methods and ideas could enhance our work, giving our approach **significant potential** for improvement. Furthermore, we are confident that there is **considerable room for performance enhancement**, as demonstrated by the improvements brought by our adjusted training strategy.
>
>
> - **Q4. The authors claim that due to resource constraints, the ablation studies are conducted on a subset of 10,000 images. This makes the evaluation not very convincing. The overall framework can be implemented on the whole dataset. The ablation study without some components can be also conducted.**
>
>   Thanks for your valuable comment. We have **expanded the training data for the ablation experiment** by 4 times, and the updated results are as follows. Due to resource limitations, we have currently used 40,000 images, which is 4 times the amount used previously, as the training data. In the future, we plan to conduct ablation experiments on the full dataset.
>
>   - Table 3: Ablation study on decoder components：
>
>     | CTI | OFA | Align | TETA           | AssocA         | ClsAb          | ClsAn         |
>     | --- | --- | --- | -- | -- | -- | --- |
>     | √  |     |       | 31.2           | 31.8           | 12.6           | 1.9           |
>     | √  | √  |       | 31.9           | 32.0           | 13.9           | 2.1           |
>     | √  |     | √    | 32.5           | 33.9           | 13.8           | 2.0           |
>     | √  | √  | √    | **33.2** | **34.5** | **15.2** | **2.7** |

---

> ### Author Response · Authors · 2024-11-27
> **Response to Reviewer tyWH (Part 4)**
>
> (Continued from Part 3)
>
>   - Table 4: Ablation study on the protection strategies for the decoders：
>
>     | Category | Content | TETA           | AssocA         | ClsAb          | ClsAn         |
>     | -- | --- | -- | -- | -- | --- |
>     |          |         | 32.1           | 32.8           | 14.6           | 2.3           |
>     | √       |         | 32.2           | 33.0           | **15.6** | 2.5           |
>     |          | √      | 32.4           | 33.6           | 14.3           | 2.5           |
>     | √       | √      | **33.2** | **34.5** | 15.2           | **2.7** |
>
>   - Table 5: Ablation study on alignment methods:
>
>     | Alignment    | TETA           | AssocA         | ClsAb          | ClsAn         |
>     | -- | -- | -- | -- | --- |
>     | Text         | 31.6           | 32.6           | 14.0           | 2.0           |
>     | Image (used) | **33.2** | **34.5** | **15.2** | 2.7           |
>     | Avg          | 32.3           | 33.2           | 14.1           | **3.2** |
>
>   - Table 6: Ablation study on inputs for CIP $I_\text{CIP}$:
>
>     | $I_\text{CIP}$ | TETA           | AssocA         | ClsAb          | ClsAn         |
>     | -- | -- | -- | -- | --- |
>     | Otxt             | 32.5           | 33.8           | 14.7           | 1.9           |
>     | Oimg(used)       | **33.2** | **34.5** | **15.2** | **2.7** |
>
>   - Table 7: Different Image Sequence Lengths for Multi-Frame Optimization:
>
>     | Length  | TETA           | AssocA         | ClsAb          | ClsAn         |
>     | --- | -- | -- | -- | --- |
>     | 2       | 27.9           | 24.9           | 13.5           | 2.2           |
>     | 3       | 30.5           | 32.2           | 14.5           | 2.5           |
>     | 4       | 31.6           | 33.8           | 14.7           | 2.4           |
>     | 5(used) | **33.2** | **34.5** | **15.2** | **2.7** |
>
>   From the results above, we observe that the ablation experiment conducted with 40,000 images yields relative performance among different ablation settings that is largely consistent with the results obtained from training on 10,000 images. This new result aligns with our previous analysis. With the increased data volume, the performance differences between models under different ablation settings have become more pronounced, highlighting the advantages of these configurations. It can still be concluded that our proposed CIP, dual-branch structure, and decoder protection strategy contribute to significant performance improvements.
>
>   **Additionally, we conducted ablation studies by removing certain components. The results are as follows.**
>
>   - **Ablation study on OVTR components:**
>
>     | CIP | Isolation | Dual | TETA           | AssocA         | ClsAb          | ClsAn         |
>     | --- | --- | -- | -- | -- | -- | --- |
>     |     |           |      | 28.9           | 29.1           | 10.7           | 1.7           |
>     | √  |           |      | 30.3           | 30.2           | 12.1           | 1.8           |
>     | √  | √        |      | 31.2           | 31.8           | 12.6           | 1.9           |
>     | √  |           | √   | 32.1           | 32.8           | 14.6           | 2.3           |
>     | √  | √        | √   | **33.2** | **34.5** | **15.2** | **2.7** |
>
>   **The first row** represents the baseline model we constructed, which **does not incorporate category information when iterating track queries—essentially, it does not establish a category information flow**. Specifically, in the single-branch structure (CTI only), we added an auxiliary network structure identical to the standard MOTR decoder, but it operates entirely independently of the CTI branch. As a result, the queries passed through this structure lack category information. When using its output as input to CIP, the category information flow lacks direct category information. **Dual** indicates the dual-branch structure, and **Isolation** represents the protective strategies within the decoder.
>
>   The results in the table show that each component effectively enhances tracking performance. OVTR achieves a 14.9% improvement on TETA compared to the constructed baseline.The adoption of the CIP strategy increases AssocA by 3.8\% and $\text{ClsA}_\text{b}$ by 13.1\%, significantly improving both association and classification. This demonstrates that our model enables harmonious collaboration between tracking and classification. Additionally, the dual-branch structure and isolation strategies further enhance the model's open-vocabulary tracking performance, aligning with the findings from our earlier ablation experiments.
>
>   We have updated the analysis of the **Ablation Study on OVTR Components** in  **Appendix B.1** .

---

> ### Author Response · Authors · 2024-11-27
> **Response to Reviewer tyWH (Part 5)**
>
> (Continued from Part 4)
>
> - **Q5. The object category in the KITTI dataset is not various, which is not very appropriate for the OVTrack task. The comparison of OVTrack and the proposed method on KITTI using Car Category is not very convective, for the OVTrack problem.**
>
>   Our experiments on the KITTI dataset aim to evaluate the generalization capability of our model. The results demonstrate that our model outperforms OVTrack in directly tracking and predicting on an untrained dataset. Moreover, our approach requires only a multi-category dataset, and the KITTI dataset provides a scene with a wide variety of category objects.
>
>   Tracking cars in this scene is particularly challenging. **While many of these categories are not included in the evaluation, they contribute to a diverse and realistic environment**. We also retained the prediction logits for multiple other categories, allowing us to track a variety of target objects in complex scenes, which makes the tracking task more difficult. This complexity **introduces actual interference when evaluating car tracking, making it a valuable test of adaptability to multi-category scenarios.** Additionally, the scarcity of open-vocabulary datasets limits our ability to conduct more extensive dataset transfer experiments.

---

> ### Comment · Reviewer_tyWH · 2024-11-28
> **Feedback after the author rebuttal**
>
> The reviewer unfeignedly thanks for the authors' effort during the rebuttal, which really needs quite some time for me to review.
>
> Above all, for my questions about the writing and performance (Q1 and Q2), the authors thoroughly updated the presentation and even the main results of the proposed method. It's hard to say... After a rushed manuscript was submitted, then the authors kept conducting and improving the experiments, which are updated for the rebuttal. The reviewers are also taken to check the detailed writing of the manuscript. Even if it's allowed, the reviewer still considers this inappropriate.
>
> Then for Q3, I acknowledge the end-to-end algorithm for the OVTrack is meaningful, whether there is an advantage or not compared to the tracking-by-detection methods.
> However, I find other reviewers also mention the writing and implementations of the proposed method. The lack of details and unclear presentation discount the significance of this work. For example, the authors say that they use image data from LVIS training and augment it to create image sequences (App C). However, I can not find how to generate the sequences.
> So, have the authors released the code and training settings, which may be more helpful than the paper?
>
> For Q4, the ablation study, it is not common to conduct the ablation study not based on the main results of the method using different data, as seen in almost all published papers.
>
> For Q5, I still think the evaluation on KITTI is not necessary, inferior to the OV dataset [R1]. The setting and goal of these two tasks are different. Compared with taking OVTrack and OVTR for comparison, the previous close-set trackers for KITTI are better. In App B.4, I can not understand why OVTR outperforms OVTrack on pedestrians by such a large margin. The CLIP-based classification learning and used annotations for pedestrians in LVIS for both methods are the same.
>
> More detailed errors.\
> In Table 2, only the scores for MT and ML are with %.\
> In Table 10, the score for FN (number of false negatives) should not be a percentage.\
> What is G-LVIS? Maybe the training data generated in OVTrack. Have the authors clarified it?\
> ...
>
> [R1] OVT-B: A New Large-Scale Benchmark for Open-Vocabulary Multi-Object Tracking, NeurIPS 2024.
>
> Overall, I acknowledge the motivation of this work. However, the paper (especially the previous version) is not ready for publication. I suggest the authors further polish it (better together with an expert), which has obtained enough comments in this submission. This is all my current comments and I am not sure about the rating. Anyway, the decision is left to AC.

---

> > ### Author Response · Authors · 2024-12-01
> > **Response to Reviewer tyWH**
> >
> > We sincerely thank Reviewer tyWH for the prompt response and for dedicating extra time to review our rebuttal revision.
> >
> > We apologize for the delay in submitting our rebuttal. This was due to the considerable amount of time we spent addressing the reviewers' concerns by adding additional experiments and method details in order to provide satisfactory responses to each question.
> >
> > **However, there are still some misunderstandings that we would like to clarify.**
> >
> > - **Q0.Revisions to the manuscript**
> >
> >   **1.** We sincerely appreciate the reviewer's continued efforts in reviewing our manuscript. Although we have made some updates to the manuscript, we believe these are in line with ICLR's unique mechanism, which encourages authors to actively improve their work based on the reviewers' suggestions. Since there are no word limits for both reviewer comments and author responses, and the rebuttal period is relatively long, this has allowed for substantial suggestions from the reviewers, as well as ample opportunity for authors to make comprehensive revisions based on these suggestions. In our rebuttal revision, we have made a genuine effort to improve the manuscript based on the suggestions provided by the reviewers.
> >
> >   **2**. To take a step back, we would like to clarify that **we did not make significant changes to the description of the method**. The **main change was adding preliminaries** for OVTR, that is, the mechanism of **MOTR**, which **may give the impression that many modifications were made** in the overview. However, this was **necessary** as it **provides the foundation for a clear explanation** of our method. We believe it greatly enhances the clarity and understanding of the manuscript. Furthermore, most of the other additions were made in response to the reviewers' valuable suggestions, such as providing more detailed descriptions of the method and adding an inference speed analysis. In addition, the **update to the experimental results** was **not** achieved by **modifying the model or method**, but rather by **simply extending the training epochs**.
> >
> >
> > - **Q1.The advantages of OVTR and implementation details (e.g., sequence generation)**
> >
> >   **1**. We would like to **thank the reviewer for acknowledging the significance of our method**. OVTR is **a novel paradigm for open-vocabulary multi-object tracking (MOT) that achieves open-vocabulary tracking in an end-to-end manner**.
> >   First, the advantages of our method, which **were specifically requested by the reviewer** to be presented, **should not be overlooked**, and they deserve further emphasis. we believe **these advantages** are even **more important than the performance scores** we reported. As **detailed in Response to Reviewer tyWH (Part 3)**, we have outlined five key advantages. **Clearly, our end-to-end framework offers a more elegant and effective solution.**
> >
> >   **2**. As the reviewer mentioned, we have received feedback from two other reviewers regarding the method details and presentation. However, it is important to note that this feedback pertains to the **previous submitted version**. Since then, we have made adjustments to both the writing and the implementation, adopting more symbolic expressions to enhance clarity. We believe that, with the revisions, the description of the method and the presentation of the experiments have been significantly improved and will now meet the expectations of the reviewers and the area chair.
> >
> >   **3**. Additionally, following the reviewers' suggestions, we had included more details in Appendix A.4 **(Isolation Strategies)**, C **(Static Image Data Augmentations)**, and D **(Model and Training Hyperparameters).**
> >
> >   **4**. In particular, regarding **the data augmentation methods in Appendix C**, we believe we have **provided a clear explanation**, including illustrative diagrams (**Fig.5**) and examples of augmentation outputs (**Fig.6**) to aid understanding.
> >
> >   **5**. Regarding the release of the code, after ICLR, we will upload the revised paper to arXiv and **make the code publicly available**. Additionally, the **training settings** are provided in Appendix D, and more details will be available in the code.
> >
> > - **Q2. Ablation experiment data volume**
> >
> >   **1**. Regarding the data size used in ablation experiments in other papers, **the comparison method, OVTrack [1]**, performs ablation experiments using **only 10,000 images**. In contrast, we believe that using **40,000 images** in our **OVTR** is already sufficiently persuasive.
> >
> >   *[1] Li, Siyuan, et al. "Ovtrack: Open-vocabulary multiple object tracking." Proceedings of the IEEE/CVF conference on computer vision and pattern recognition. 2023.*

---

> > ### Author Response · Authors · 2024-12-01
> > **Response to Reviewer tyWH**
> >
> > **2**. This is because the OVMOT model augments each image into multiple frames during training, meaning the actual number of images used per epoch is **much greater than** the total number of static images (**40,000**). Due to our **extensive and detailed ablation experiments (15 ablation settings)**, the resource consumption during training is considerable, when using 40,000 static images. (In contrast, the entire MOT20 dataset only contains a total of 8,931 training frames.)
> >
> >   **3**. In fact, we have increased the data size for our ablation experiments from 10,000 to 40,000, which is nearly **half of the training data** used in performance comparison experiments. Moreover, we have analyzed the results from both 10,000 and 40,000 image datasets, and **the conclusions drawn from these two settings are consistent**. This provides sufficient evidence that our ablation experiments are persuasive.
> >
> > - **Q3. The evaluation on other datasets**
> >
> >   **1**. First, following the reviewer's suggestion, we evaluate our method on the OVT-B dataset (from **concurrent work**), and the results are as follows:
> >
> >     | **OVT-B** | Data         | ${\text{Proposals}}_{\text{novel}}$ | End-to-End   | TETA↑(novel)  | LocA↑(novel)  | AssocA↑(novel) | ClsA↑(novel) | TETA↑(base)   | LocA↑(base)   | AssocA↑(base) | ClsA↑(base)   |
> >     | -- | -- | --- | -- | -- | --- | --- | -- | -- | -- | -- | -- |
> >     | OVTrack                  | G-LVIS, LVIS | ✓                     |               | 45.5           | 61.1           | 65.5            | **9.6**           | 46.8           | 60.5           | 66.7           | **13.4** |
> >     | OVTrack+(from OVT-B, **concurrent**)                 | G-LVIS, LVIS | ✓                    |                | **46.4**           | **62.5**           | 67.3            | 9.4           | **47.6**           | **61.6**           | 68.2           | 13.2 |
> >     | OVTR                     | LVIS    |   |      **✓**     | 45.5 | 59.7  |  **67.6**    |  9.3   |  **47.6**    |  60.9   | **68.9** |  12.9 |
> >
> >   The results clearly show that OVTR surpasses OVTrack on base TETA, matching the performance of OVTrack+ (an improved version of OVTrack, a concurrent work to ours). Even without using proposals containing novel categories, OVTR achieves similar TETA results to OVTrack on novel categories, demonstrating the effectiveness of our model design. Additionally, our method achieves the highest AssocA for both base and novel categories, indicating superior tracking performance in high-frame-rate video (OVT-B). Overall, our approach outperforms OVTrack.
> >
> >   We will include the evaluation results on OVT-B in the camera-ready version. The code for **evaluating OVTR on OVT-B will also be released** to the public.
> >
> >   **2**. We use the KITTI dataset to validate whether the OVMOT model can exhibit tracking capabilities in specialized traffic scenarios. We aim to apply **a unified tracking algorithm across different environments**. For example, **switching** between different closed-set trackers in various scenes such as in wilderness areas, in traffic, during races, or from aerial views **is not a wise decision**. Additionally, **many complex categories in traffic scenarios cannot be recognized by these closed-set trackers** or require additional training, such as road signs, traffic barriers, taxis, phone booths, and charging stations.
> >
> >   **3**. Regarding the **results** of OVTR **on pedestrians**, it outperforms OVTrack significantly, which **matches the actual output results tested on the TAO dataset**. Examples of this can be seen in Figure 9 in Appendix E, where **OVTrack fails to recognize the "person" category throughout the entire phase**, whereas **we are able to maintain long-term tracking of the "person" category.**
> >
> >   - Our explanation for this is as follows:
> >
> >   **Firstly, we would like to reiterate, as mentioned in Appendix A, that our classification learning differs from OVTrack.** Specifically, during training, OVTR samples 250 category texts as input, to engage in information interaction and contrastive learning.
> >   We attribute OVTrack's **poor performance on the "pedestrian" category** to the **incomplete labeling** of the "person" category in the LVIS dataset. During OVTrack training, the logits for all 1,203 categories are involved in the loss calculation. However, due to incomplete annotations for the 'person' category, this causes the 'person' to be gradually misclassified as the background category, suppressing its recognition. In contrast, during training, we sample a total of 250 categories (often excluding the 'person' category, reducing the model's learning of 'person' classification), to compute logits. This originally aimed to suppress the long-tail distribution but now also prevents the suppression of the "person" category. (Note that OVTR inference evaluation involves 1203 categories.) The details of how we sample category texts can be found in Appendix A.1.

---

> > ### Author Response · Authors · 2024-12-01
> > **Response to Reviewer tyWH**
> >
> > - **Q4. Detail issues**
> >   To maintain consistency, we have removed the % in Table 2. Additionally, the value for FN in Table 10 represents the FN ratio, in order to maintain consistency with the table format used in the comparison method CenterTrack [2], as shown in Table 4 of [2]. Following the reviewer's suggestion, we have now updated the FN value in Table 10. Furthermore, we have added explanation of G-LVIS in Table 1.
> >
> >     *[2] Zhou, Xingyi, Vladlen Koltun, and Philipp Krähenbühl. "Tracking objects as points." European conference on computer vision. Cham: Springer International Publishing, 2020.*
> >
> > We appreciate the reviewer's recognition of the motivation behind our work. OVTR is a novel paradigm of open-vocabulary MOT that achieves OV tracking in an end-to-end manner. This approach **offers** the OV-MOT community **a new research direction** to explore. We believe that **many new methods and ideas could enhance our work**, offering **substantial potential for improvement**. Furthermore, we are confident that there is **considerable room for performance enhancement**, as demonstrated by the improvements brought by our adjusted training strategy.
> >
> > We would like to emphasize once again that, as shown in **Response to Reviewer tyWH (Part 3)**, we have highlighted five advantages that are independent of the evaluation results. Aside from the performance improvements, these are also **the core advantages of our method**.
> >
> > We sincerely thank the reviewer once again for his/her additional time and effort in providing thoughtful feedback.
> >
> > Since ICLR allows authors to revise the manuscript based on constructive feedback from reviewers, it encourages authors to incorporate and improve upon these suggestions. We kindly ask the reviewer to consider our updated rebuttal revision, as it better reflects our genuine effort to address the reviewers' valuable suggestions.

---

> ### Comment · Reviewer_tyWH · 2024-12-02
> **Reviewer feedback**
>
> The reviewer unfeignedly thank the authors for their efforts again. The rebuttal is much better than the manuscript (submitted version).
>
> I would like to raise my score considering the improvement of the current version and the persuasive response.
>
> But I still strongly suggest the authors submit a mature version next time for the next work, even if to ICLR. This may gain a higher initial and/or final score for you after the rebuttal.

---

> > ### Author Response · Authors · 2024-12-03
> >
> > We are pleased that our response has been considered persuasive by the reviewer.
> > In future work, we will strive to ensure greater maturity in the initial submission.
> >
> > Once again, we sincerely appreciate the reviewer for dedicating additional time to review our revisions.

---

### Official Review · Reviewer_dQGj · 2024-11-07

**Soundness:** 3
**Presentation:** 2
**Contribution:** 4
**Rating:** 8
**Confidence:** 5

**Summary:**

This paper proposes the first solution for end-to-end open-vocabulary multiple object tracking called OVTR. The model builds on close-set end-to-end multiple object tracker MOTR while extending it to open-vocabulary by adding the category Information Propagation strategy and dual-branch vision-language alignment training. The method achieves good results on open-vocabulary tracking benchmarks and also demonstrates good zero-shot transfer ability to unseen domains.

**Strengths:**

1. The paper presents the first end-to-end Transformer solution for open-vocabulary tracking that performs joint detection and tracking. This is known to be a hard problem for two main reasons:
   - There is a lack of sufficiently diverse video data to train end-to-end models effectively. Training on a small set of videos for tracking often hurts detection performance. The paper addresses this by augmenting static images for training, demonstrating that this approach can still yield good performance in an end-to-end setup. Although it is not new to use static images for tracking it is rare in end-to-end solutions, especially in open-vocabulary scenarios.
   - In end-to-end multi-object tracking solutions, queries often perform multiple tasks—tracking, localization, and classification—which is really difficult to balance in open-vocabulary scenarios. The paper’s proposed category and content isolation strategies help balance these tasks, which is a sensible approach.

2. The zero-shot transfer experiments on the KITTI dataset indicate strong performance in association, particularly in terms of the IDF1 metric, showing significant improvement over previous methods like OVTrack.

**Weaknesses:**

1. The paper is not clearly written and lacks important details. For instance, understanding the overall framework and its training process is challenging. The paper mentions using only static images to train the framework; however, using static images implies there is no separation of new and old objects, which are typically handled differently in MOTR training. This critical aspect is not clearly explained in the paper.

2. The ablation results in Table 3 and Table 4 are confusing and show only marginal improvements over the baseline, especially in association. For example, in Table 4, adding category isolation only improves ClsA_b by 0.6 and ClsA_n by 0.1, while adding content isolation alone only improves 0.1 in AssocA performance. A more detailed explanation is needed here to clarify these results.​

3. Missing latest related works. The paper only compares OVTrack. However, In open-vocabulary tracking, there are some new advancements eg. MASA[1] at CVPR2024. Also, SLAck at ECCV2024, which is also a transformer-based end-to-end approach. Maybe SLAck can be a concurrent work but MASA seems to be fair to discuss.

[1] Li, Siyuan, et al. "Matching Anything by Segmenting Anything." Proceedings of the IEEE/CVF Conference on Computer Vision and Pattern Recognition. 2024.

**Questions:**

Will the code be public? Since this is the first work that performs end-to-end open-vocabulary object tracking, making the implementation details public is of great importance to the research community.

---

> ### Comment · Reviewer_dQGj · 2024-11-25
>
> After reading others’ reviews and noting the lack of response from the authors, I have decided to downgrade my rating. However, I still believe that being the first work to study open-vocabulary in an end-to-end fashion holds unique value for the research community. The key issue lies in the presentation, which is very unclear in this paper.

---

> > ### Author Response · Authors · 2024-11-27
> >
> > We sincerely apologize for not responding to you earlier. The reason for the delay is that the experiments and explanations required were extensive. We have diligently completed the supplementary experiments and provided thorough explanations. We greatly appreciate your valuable feedback, and we have made every effort to address your questions as comprehensively as possible.

---

> ### Author Response · Authors · 2024-11-27
> **Response to Reviewer dQGj (Part1)**
>
> We sincerely appreciate your thorough review of our paper and the valuable insights you provided.
>
> - **Q1. The paper is not clearly written and lacks important details. For instance, understanding the overall framework and its training process is challenging.**
>
>   Thanks for your valuable comment. Considering your suggestions, we have updated the content of our manuscript. For specific improvements, please refer to point 2 in the common response. Here, we provide an overall explanation of the approach and model. Additionally, we have made adjustments in the updated rebuttal revision, addressing areas where the original method section was unclear. The revised parts are highlighted in red for easy identification.
>
>   **The overall framework and training process:**
>
>   Our model is illustrated in **Figure 2 of the updated manuscript** (with added details). **We follow the general structure and basic tracking mechanism of MOTR, treating MOT as an iterative sequence prediction problem.** **Each trajectory is represented by a track query**. (**The basic structure and mechanisms related to MOTR are provided in Sec.3.1 of the rebuttal revision, beginning from line 158**)
>
>   Let us first revisit the representative iterative query-based tracker, MOTR. Following a DETR-like structure, detect queries  $Q_\text{det}^{t=1}$  for the first frame $f_{t=1}$ are fed into the Transformer decoder, where they interact with the image features $E_\text{img}^{t=1}$ extracted by the Transformer encoder. This process yields updated detect queries $Q_\text{det}^{\prime\,{t=1}}$ that contain object information. Detection predictions, including bounding boxes $B_\text{det}^{t=1}$ and object representations $O_\text{det}^{t=1}$, are subsequently extracted from $Q_\text{det}^{\prime\,{t=1}}$.
>
>   In contrast to DETR, for the query-based iterative tracker, $Q_\text{det}^{t=1}$ are only needed to detect newly appeared objects in the current frame. Consequently, one-to-one assignment is performed through bipartite matching exclusively between $Q_\text{det}^{\prime\,{t=1}}$ and the ground truth of the newly appeared objects, rather than matching with the ground truth of all objects.
>
>   The matched $Q_\text{det}^{\prime\,{t=1}}$ will be used to update and generate the track queries $Q_\text{tr}^{t=2}$, which, for the second frame $f_{t=2}$, are fed once again into the Transformer decoder and interact with the image features $E_\text{img}^{t=2}$ to obtain the representations and locations of the objects matched with $Q_\text{tr}^{ t=2}$, thereby enabling tracking predictions. The $Q_\text{tr}^{t=2}$ maintain their object associations and are updated to generate the $Q_\text{tr}^{t=3}$ for the third frame $f_{t=3}$. Parallel to $Q_\text{tr}^{t=2}$, and similar to the process for $f_{t=1}$, $Q_\text{det}^{t=2}$ are fed into the decoder to detect newly appeared objects. $Q_\text{det}^{\prime\,{t=2}}$ undergo binary matching, and the matched queries are transformed into new track queries, which are then added to $Q_\text{tr}^{t=3}$ for $f_{t=3}$. The entire tracking process can be extended to subsequent frames.
>
>   Regarding optimization, MOTR employs multi-frame optimization, where the loss is computed by considering both ground truths and matching outcomes. The matching results for each frame include both the maintained track associations and the binary matching results between $Q_\text{det}^\prime $ and newly appeared objects.
>
>   During inference: Similar to MOTR, the network forward process during inference in OVTR follows the same procedure as during training. The key difference lies in the conversion of track queries. In detection predictions, if the category confidence score exceeds $\tau_\text{det}$, the corresponding updated detect query is transformed into a new track query, initiating a new track. Conversely, if a tracked object is lost in the current frame (confidence $\leq \tau_\text{tr}$), it is marked as an inactive track. If an inactive track is lost for $T_\text{miss}$ consecutive frames, it is completely removed.

---

> ### Author Response · Authors · 2024-11-27
> **Response to Reviewer dQGj (Part 2)**
>
> (Continued from Part 1)
>
>  Leveraging the iterative nature of the query-based framework, OVTR transfers information about tracked objects across frames, aggregating category information throughout continuous image sequences to achieve robust classification performance, rather than performing independent localization and classification in each frame.
>
>   In the encoder, preliminary image features from the backbone and text embeddings from the CLIP model are processed through pre-fusion to generate fused image features $E_{img}$ and text features $E_{txt}$. We propose a dual-branch decoder comprising the OFA branch and the CTI branch. Upon input of $Q = [Q_\text{det}, Q_\text{tr}]$, the two branches respectively guide $Q$ to derive visual generalization representations and perform deep modality interaction with $E_\text{txt}$, outputting $O_\text{img}$, $O_\text{txt}$. $O_\text{img}$ serve as the input for the category information propagation (CIP) strategy, injecting category information into the category information flow. This process is an extension of the aforementioned mechanism where $Q_\text{det}^{\prime\,t}$ generates $Q_\text{tr}^{t+1}$. Meanwhile, $O_\text{txt}$ are utilized for computing category logits and for contrastive learning.
>
>   **The details in sections 3.2, 3.3, 3.4, and 3.5 have been supplemented to provide a clearer and more understandable explanation of the method. Changes are visible in red font.**
>
>   We hope that within the limited space here, we have provided as clear an explanation of our work as possible. More details can be found in the updated method section, marked in red, and in the supplementary materials (**Appendix A C D** of the rebuttal revision).
>
> - **Q2. The paper mentions using only static images to train the framework; however, using static images implies there is no separation of new and old objects, which are typically handled differently in MOTR training. This critical aspect is not clearly explained in the paper.**
>
>   Thanks for your valuable comment. We will provide a detailed explanation from the following aspects.
>
>   **Static Image Data Augmentation:** As noted by the reviewers, MOTR-like query-based trackers require a different approach to data augmentation. These trackers have higher demands for training data. Our data augmentation includes conventional techniques, such as applying random resizing, horizontal flipping, color jittering, and random affine transformations to single images to create distinguishable multi-frame data. This part aligns with OVTrack. Additionally, we propose **Dynamic Mosaic** and **Random Occlusion** augmentations, specifically designed for MOTR-like trackers. Detailed explanations, illustrations, and examples of these methods have been added to the supplementary materials in the updated rebuttal revision.
>
>   Specifically, while query-based trackers excel at maintaining associations over extended periods, they place higher demands on the realism of object motion patterns in training data. For OVTR, track queries must not only learn to capture the same object as it moves to a new position in the next frame, but also handle scenarios such as **object appearance** and  **disappearance** ,  **tracking continuity after occlusion** , and maintaining correct associations when  **relative motion occurs between tracked objects and others** .
>
>   To address these challenges, we propose **Dynamic Mosaic** augmentation, an improvement over the Mosaic augmentation in OVTrack. In addition to stitching four different images into a single composite, Dynamic Mosaic generates images with varying relative spatial relationships among objects across different training frames. This simulates scenarios such as objects approaching or receding from each other, crossing paths, and exhibiting relative size changes. The **Random Occlusion** augmentation is employed to simulate situations where objects disappear due to occlusion and then reappear or suddenly emerge in the scene.
>
>   More detailed augmentation techniques and illustrative examples can be found in the updated supplemental materials **Appendix C** of the rebuttal revision, **beginning from line 805**.
>
>
> - **Q3. The ablation results in Table 3 and Table 4 are confusing and show only marginal improvements over the baseline, especially in association. A more detailed explanation is needed here to clarify these results.**
>
>   The differences between the results in Tables 3 and 4 are relatively minor, which we believe is mainly due to the following **reasons**:
>
>   1. The limited training data size used in the ablation studies constrained the model's performance, making it difficult to demonstrate significant differences.
>   2. The ablation experiments focused on the internal modifications of each module, where the relative improvements were less pronounced.

---

> > ### Comment · Reviewer_dQGj · 2024-11-28
> >
> > Thank you for adding the detailed explanations regarding the method and the static data augmentation. I appreciate the authors’ efforts.
> > The proposed method is based on an MOTR-style tracker, which uses various cues such as location, appearance, and classes to make tracking decisions. One remaining concern is that the augmentation created from static images lacks realistic motion. Training with such unrealistic motion could introduce a significant gap between training and inference. For example, as shown in Fig. 5 of the supplementary material, T2 and T3 employ dynamic mosaic augmentation, which exchanges the locations of sub-images, resulting in large motion displacements. However, this scenario is rarely encountered in real videos, where objects typically exhibit much smaller motion displacements between consecutive frames. OVTrack is purely an appearance-based tracker, so employing such augmentation is understandable. However, for trackers that also consider motion cues, wouldn’t this pose a serious issue?

---

> > > ### Author Response · Authors · 2024-12-01
> > > **Response to Reviewer dQGj**
> > >
> > > We sincerely appreciate Reviewer dQGj for the prompt response and valuable suggestions.
> > >
> > > - **Q1. Effectiveness of data augmentation**
> > >
> > >   In fact, we considered the concern raised by the reviewer during the design of our data augmentation strategies, as query-based trackers require training data with more realistic motion.
> > >
> > >   **1.** First, Fig. 5 is designed to highlight the effects and make them clearer for the readers. However, in practice, the Dynamic Mosaic augmentation **does not cause excessive motion displacement or size changes**.
> > >
> > >   **2.** Additionally, after Dynamic Mosaic augmentation and Random Occlusion augmentation, the multi-frame images are **sequentially translated in one of eight directions**—up, down, left, right, upper-left, upper-right, lower-left, or lower-right—following general motion patterns.
> > >
> > >   **3.** Furthermore, **the swapping of sub-image** positions, which we found necessary through preliminary experiments, especially improves tracking performance in situations where **an object is occluded by another and then reappears from the other side**. This augmentation also **introduces some irregular motion**, which is particularly beneficial for tracking.
> > >
> > >   **4.** In fact, based on our output results, **our tracking performance is strong** when the **motion displacement** between consecutive frames **is small**. We believe this is due to two reasons:
> > >
> > >     - Query-based tracking relies on track queries to locate objects in the current frame. When the motion displacement is small, **the appearance and location of the object in the current frame is similar to that in the previous frame**. As a result, track queries that retain object information (both location and appearance) from the previous frame are naturally **more likely to locate the object** in the current frame.
> > >
> > >     - Some image sequences are augmented with larger motion displacement than those seen during inference. In fact, this does not cause query-based tracking to overfit to handling greater motion displacement. On the contrary, **it enhances the robustness of the query-based tracker.**
> > >
> > >   **5.** From an evaluation perspective, the majority of the motion displacement of objects in open-vocabulary tracking scenarios is large, due to many non-professional, unstable captures, and the frames being sampled at 30-frame intervals. While the motion follows general patterns, the changes in object position and size remain significant.
> > >
> > >   **6.** We also conducted additional tests on the OVT-B dataset from concurrent work, **where the object motion displacement is relatively small** due to the high frame rate of the annotated video. The evaluation results are as follows. (Note that MASA is not included here, but it will be added for comparison in the main table.)
> > >     | **OVT-B** | Data         | ${\text{Proposals}}_{\text{novel}}$ | End-to-End   | TETA↑(novel)  | LocA↑(novel)  | AssocA↑(novel) | ClsA↑(novel) | TETA↑(base)   | LocA↑(base)   | AssocA↑(base) | ClsA↑(base)   |
> > >     | -- | -- | --- | -- | -- | --- | --- | -- | -- | -- | -- | -- |
> > >     | OVTrack                  | G-LVIS, LVIS | ✓                     |               | **45.5**           | **61.1**           | 65.5            | **9.6**           | 46.8           | 60.5           | 66.7           | **13.4** |
> > >     | OVTR                     | LVIS    |   |      **✓**     | **45.5** | 59.7  |  **67.6**    |  9.3   |  **47.6**    |  **60.9**   | **68.9** |  12.9 |
> > >
> > >   The results clearly show that our method achieves higher AssocA for both base and novel categories, indicating superior tracking performance in high-frame-rate video (OVT-B). This demonstrates the effectiveness of our data augmentation techniques, which enable the model to **effectively track objects with small motion displacement**.
> > >
> > >   Overall, compared to the extent of translation and position swapping, the relative position or size changes introduced by **Dynamic Mosaic**, as well as the simulation of appearance and disappearance through **Random Occlusion**, are **more important** for query-based methods.

---

> ### Author Response · Authors · 2024-11-27
> **Response to Reviewer dQGj (Part 3)**
>
> (Continued from Part 2)
>
>   To address these issues, we conducted experiments using a dataset **4** times larger (40,000 images) than that of the previous ablation studies and supplemented the results with  **ablation studies without some components** . Details are as follows.
>
>   - Table 3: Ablation study on decoder components：
>
>     | CTI | OFA | Align | TETA           | AssocA         | ClsAb          | ClsAn         |
>     | --- | --- | ----- | -------------- | -------------- | -------------- | ------------- |
>     | √  |     |       | 31.2           | 31.8           | 12.6           | 1.9           |
>     | √  | √  |       | 31.9           | 32.0           | 13.9           | 2.1           |
>     | √  |     | √    | 32.5           | 33.9           | 13.8           | 2.0           |
>     | √  | √  | √    | **33.2** | **34.5** | **15.2** | **2.7** |
>
>   - Table 4: Ablation study on the protection strategies for the decoders：
>
>     | Category | Content | TETA           | AssocA         | ClsAb          | ClsAn         |
>     | -------- | ------- | -------------- | -------------- | -------------- | ------------- |
>     |          |         | 32.1           | 32.8           | 14.6           | 2.3           |
>     | √       |         | 32.2           | 33.0           | **15.6** | 2.5           |
>     |          | √      | 32.4           | 33.6           | 14.3           | 2.5           |
>     | √       | √      | **33.2** | **34.5** | 15.2           | **2.7** |
>
>   - Table 5: Ablation study on alignment methods:
>
>     | Alignment    | TETA           | AssocA         | ClsAb          | ClsAn         |
>     | ------------ | -------------- | -------------- | -------------- | ------------- |
>     | Text         | 31.6           | 32.6           | 14.0           | 2.0           |
>     | Image (used) | **33.2** | **34.5** | **15.2** | 2.7           |
>     | Avg          | 32.3           | 33.2           | 14.1           | **3.2** |
>
>   - Table 6: Ablation study on inputs for CIP $I_\text{CIP}$:
>
>     | $I_\text{CIP}$ | TETA           | AssocA         | ClsAb          | ClsAn         |
>     | ---------------- | -------------- | -------------- | -------------- | ------------- |
>     | Otxt             | 32.5           | 33.8           | 14.7           | 1.9           |
>     | Oimg(used)       | **33.2** | **34.5** | **15.2** | **2.7** |
>
>   - Table 7: Different Image Sequence Lengths for Multi-Frame Optimization:
>
>     | Length  | TETA           | AssocA         | ClsAb          | ClsAn         |
>     | ------- | -------------- | -------------- | -------------- | ------------- |
>     | 2       | 27.9           | 24.9           | 13.5           | 2.2           |
>     | 3       | 30.5           | 32.2           | 14.5           | 2.5           |
>     | 4       | 31.6           | 33.8           | 14.7           | 2.4           |
>     | 5(used) | **33.2** | **34.5** | **15.2** | **2.7** |
>
>   From the results above, we observe that the ablation experiment conducted with 40,000 images yields relative performance among different ablation settings that is largely consistent with the results obtained from training on 10,000 images. This new result aligns with our previous analysis. With the increased data volume, **the performance differences between models under different ablation settings have become more pronounced**, highlighting the advantages of these configurations and the potential for **further scaling up OVTR**. It can still be concluded that our proposed CIP, dual-branch structure, and decoder protection strategy contribute to significant performance improvements.

---

> ### Author Response · Authors · 2024-11-27
> **Response to Reviewer dQGj (Part 4)**
>
> (Continued from Part 3)
>
>   Additionally, to verify whether each module of OVTR contributes to performance improvement, **we conducted ablation studies by removing certain components.** The results are as follows.
>
>   - **Ablation study on OVTR components:**
>
>     | CIP | Isolation | Dual | TETA           | AssocA         | ClsAb          | ClsAn         |
>     | --- | --------- | ---- | -------------- | -------------- | -------------- | ------------- |
>     |     |           |      | 28.9           | 29.1           | 10.7           | 1.7           |
>     | √  |           |      | 30.3           | 30.2           | 12.1           | 1.8           |
>     | √  | √        |      | 31.2           | 31.8           | 12.6           | 1.9           |
>     | √  |           | √   | 32.1           | 32.8           | 14.6           | 2.3           |
>     | √  | √        | √   | **33.2** | **34.5** | **15.2** | **2.7** |
>
>   **The first row** represents the baseline model we constructed, which **does not incorporate category information when iterating track queries—essentially, it does not establish a category information flow**. Specifically, in the single-branch structure (CTI only), we added an auxiliary network structure identical to the standard MOTR decoder, but it operates entirely independently of the CTI branch. As a result, the queries passed through this structure lack category information. When using its output as input to CIP, the category information flow lacks direct category information. **Dual** indicates the dual-branch structure, and **Isolation** represents the protective strategies within the decoder.
>
>   **The results** in the table show that **each component effectively enhances tracking performance.** OVTR achieves a 14.9% improvement on TETA compared to the constructed baseline.The adoption of the CIP strategy increases AssocA by 3.8\% and $\text{ClsA}_\text{b}$ by 13.1\%, significantly improving both association and classification. This demonstrates that our model enables harmonious collaboration between tracking and classification. Additionally, the dual-branch structure and isolation strategies further enhance the model's open-vocabulary tracking performance, aligning with the findings from our earlier ablation experiments.
>
>   We have updated the analysis of the **Ablation Study on OVTR Components** in  **Appendix B.1** .
>
> - **Q4. Missing latest related works**
>
>   - MASA and SLAck are both valuable works. For the concurrent work SLAck, it is not a transformer-based end-to-end approach. As mentioned in Section 4.3 of the paper, it “employs the same Faster R-CNN detector as OVTrack,” indicating that this model is a track-by-detection tracker based on an off-the-shelf open-vocabulary detector. In contrast, our work introduces a novel transformer-based end-to-end structure. Regarding MASA, we believe its approach is not directly comparable to ours. MASA distills knowledge from SAM and combines it with off-the-shelf open-vocabulary detectors for inference. Strictly speaking, the foundational conditions of our method differ significantly from those of MASA.
>   - First, we adjusted the model's training strategy,  **leading to enhanced performance**. Specifically, the number of training epochs was increased from 12 to **15**. At the 13th epoch, the learning rate underwent an additional decay by a factor of 10. Multi-frame training was adjusted, increasing the number of frames from 2 to 3, 4, and **5** at the 4th, 7th, and **14th** epochs, respectively (previously 2, 3, and 4 at the 4th and 7th epochs). This allowed the model to accurately locate and track objects over longer video sequences. The performance of the model trained with this strategy is presented in the table below. The results demonstrate that, compared to OVTrack, our method achieved a **12.9%** improvement on the TETA metric for novel categories and a **3.1%** improvement for base categories on the validation set. On the test set, TETA increased by **12.4%** for novel categories and **5.8%** for base categories.

---

> > ### Comment · Reviewer_dQGj · 2024-11-28
> >
> > For Q3, I appreciate the authors' efforts to conduct the ablation on a larger scale. The explanation makes sense to me.
> >
> > For Q4, I appreciate the authors’ explanation and detailed comparison with the latest works. I would suggest adding the results—for example, comparing to MASA—in the main table, as the current comparison includes only one recent tracker, OVTrack. Additionally, I appreciate the discussion regarding SLAck, which uses an end-to-end transformer for association instead of joint association and detection. I suggest updating the related work section to include these discussions, as it would help readers better understand the differences between the proposed method and other published works.
> >
> > I also appreciate the authors' promise to public the code.

---

> > > ### Author Response · Authors · 2024-12-01
> > > **Response to Reviewer dQGj**
> > >
> > > - **Q2. Latest works**
> > >
> > >   We would like to thank the reviewer for the valuable suggestions. Based on the reviewer's feedback, we have decided to include a comparison with MASA in our main table in the next version (e.g., camera-ready). We will also include both SLAck and MASA in the Related Work section to discuss the differences between OVTR and these methods.
> > >
> > >   Additionally, we would like to emphasize that we did not use any external OV-detector; our OVTR is **the first end-to-end transformer-based open-vocabulary tracker.**
> > >
> > >   Finally, we would like to **reiterate the following advantages of OVTR**:
> > >
> > >   **1.** Our method is **end-to-end**, **eliminating the need for complex post-processing and explicit similarity associations between frames**. Our model achieves faster inference speeds compared to OVTrack.
> > >
> > >     We evaluated the inference speed of OVTrack and OVTR on a single NVIDIA GeForce RTX 3090 GPU. The results, summarized in the table below, indicate that OVTR achieves faster inference compared to OVTrack. Additionally, we tested a lightweight version, OVTR-Lite, which excludes the category isolation strategy and tensor KL divergence computation. Despite some performance trade-offs, OVTR-Lite still outperforms OVTrack in overall performance. It achieves nearly **4× faster inference speed** while maintaining GPU memory usage within **4GB** during inference.
> > >
> > >     | **Methods**   | **FPS**  | TETA↑(novel)  | LocA↑(novel)  | AssocA↑(novel) | ClsA↑(novel) | TETA↑(base)   | LocA↑(base)   | AssocA↑(base) | ClsA↑(base)   |
> > >     | --- | -- | -- | -- | --- | --- | -- | -- | -- | -- |
> > >     | OVTrack             | 3.1            | 27.8           | 48.8           | 33.6            | 1.5           | 35.5           | 49.3           | 36.9           | **20.2** |
> > >     | OVTR                | **3.4**  | **31.4** | **54.4** | **34.5**  | **5.4** | **36.6** | **52.2** | **37.6** | 20.1           |
> > >     | **OVTR-Lite** | **12.4** | 30.1           | 52.7           | 34.4            | 3.14          | 35.6           | 51.3           | 37.0           | 18.6           |
> > >
> > >     The speed test experiments demonstrating the inference speed of our model have been added in Appendix B.3 of our rebuttal revision, beginning from line 776.
> > >
> > >     **2.** Our end-to-end structure differs from OVTrack, which requires off-the-shelf OVDs and the adjustment and training of two separate models. This makes our approach **more convenient** for use across diverse scenarios, without the need for adjusting two models.
> > >
> > >     **3.** Our method **does not require a separate RPN-based detector to pre-extract proposals** (which may contain novel class information, but our method does not rely on it). Additionally, our OVTR does not require using the CLIP image encoder to generate embeddings for these proposals, which can result in hundreds of gigabytes of embeddings. This significantly **saves time and reduces operations** during the preprocessing stage.
> > >
> > >     **4.** In implementing open-vocabulary tasks, our method does not rely on the presence of a large number of unannotated novel category objects in the training dataset, reducing the requirements on the dataset. Furthermore, Our method does not rely on the diffusion model mentioned in OVTrack.
> > >
> > >     **5.** OVTR is a transformer-based tracker with a data-friendly nature that facilitates scalability and possesses room for improvement.
> > >
> > >     In summary, we propose **a novel paradigm of open-vocabulary MOT** that achieves OV tracking in an end-to-end manner, introducing **a fresh perspective** on end-to-end processing and modality interaction fusion in the OVMOT field. This approach **offers** the OV-MOT community **a new research direction** to explore. We believe that **many new methods and ideas could enhance our work**, offering **substantial potential for improvement**. Furthermore, we are confident that there is **considerable room for performance enhancement**, as demonstrated by the improvements brought by our adjusted training strategy.
> > >
> > > We sincerely appreciate the reviewer’s recognition of the efforts we put into our rebuttal.
> > >
> > > Once again, we sincerely thank the reviewer for the valuable suggestions. We greatly appreciate the reviewer's insightful understanding of the OVMOT task we are addressing. We hope that our response, along with the additional clarifications, will help the reviewer better understand the significance of our work.

---

> > > > ### Comment · Reviewer_dQGj · 2024-12-02
> > > >
> > > > Thank the authors for this detailed reply! It solves most of my concerns, hence I decided to raise my score.

---

> > > > > ### Author Response · Authors · 2024-12-03
> > > > >
> > > > > We greatly appreciate the reviewer's insightful understanding of the OVMOT task, as well as the valuable suggestions provided.
> > > > >
> > > > > Once again, we are grateful for the reviewer's time and effort dedicated to reviewing our submission.

---

> ### Author Response · Authors · 2024-11-27
> **Response to Reviewer dQGj (Part 5)**
>
> (Continued from Part 4)
>
> | **Validation set** | Data         | ${\text{Proposals}}_{\text{novel}}$ | TETA↑(novel)  | LocA↑(novel)  | AssocA↑(novel) | ClsA↑(novel) | TETA↑(base)   | LocA↑(base)   | AssocA↑(base) | ClsA↑(base)   |
> | -- | -- | --- | -- | -- | --- | --- | -- | -- | -- | -- |
> | OVTrack                  | G-LVIS, LVIS | ✓                                    | 27.8           | 48.8           | 33.6            | 1.5           | 35.5           | 49.3           | 36.9           | **20.2** |
> | OVTR                     | LVIS         |                                       | **31.4** | **54.4** | **34.5**  | **5.4** | **36.6** | **52.2** | **37.6** | 20.1           |
>
> | **Test set** | Data         | ${\text{Proposals}}_{\text{novel}}$ | TETA↑(novel)  | LocA↑(novel)  | AssocA↑(novel) | ClsA↑(novel) | TETA↑(base)   | LocA↑(base)   | AssocA↑(base) | ClsA↑(base)   |
> | -- | -- | --- | -- | -- | --- | --- | -- | -- | -- | -- |
> | OVTrack            | G-LVIS, LVIS | ✓                                    | 24.1           | 41.8           | 28.7            | 1.8           | 32.6           | 45.6           | 35.4           | **16.9** |
> | OVTR               | LVIS         |                                       | **27.1** | **47.1** | **32.1**  | **2.1** | **34.5** | **51.1** | **37.5** | 14.9           |
>
> Based on the evaluation results of MASA reported in the concurrent work SLAck:
>
> | **Validation set** | Method                     | TETA↑(novel)  | LocA↑(novel)  | AssocA↑(novel) | ClsA↑(novel) | TETA↑(base)   | LocA↑(base)   | AssocA↑(base) | ClsA↑(base)   |
> | -- | -- | -- | -- | -- | --- | -- | -- | -- | -- |
> | MASA                     | **Distill-SAM**, OVD | 30.0           | 54.2           | **34.6** | 1.0           | **36.9** | **55.1** | 36.4           | 19.3           |
> | OVTR                     | End-to-End                 | **31.4** | **54.4** | 34.5           | **5.4** | 36.6           | 52.2           | **37.6** | **20.1** |
>
> **Distill-SAM** refers to knowledge distillation from the SAM model, **OVD** denotes the use of off-the-shelf open-vocabulary detectors. Even with knowledge distillation from SAM, our OVTR remains competitive with MASA in terms of localization and association. Moreover, it performs particularly well in classification, especially for novel categories.
>
>   We want to emphasize that we propose a novel open-vocabulary framework, aiming to **bring a new perspective** of end-to-end and modality interaction fusion to the OVMOT field. At the same time, we believe that many new methods and ideas could enhance our work, giving our approach **significant potential** for improvement. Furthermore, we are confident that there is considerable **room for performance enhancement**, as demonstrated by the improvements brought by our adjusted training strategy.
>
>   Even compared to MASA, which utilizes knowledge distillation from SAM, our framework still offers the following advantages:
>
>   1. First, our approach is end-to-end, eliminating the need for complex post-processing and explicit similarity association.
>   2. MASA requires an additional off-the-shelf OVD, necessitating the training and fine-tuning of two models, which becomes increasingly complex, especially in the context of diverse open-vocabulary scenarios.
>   3. (Compared to the used off-the-shelf OVD) our method **does not require a separate RPN-based detector to pre-extract proposals** (which may contain novel class information, but our method does not rely on it). Additionally, it does not require using the CLIP image encoder to generate embeddings for proposals, which can result in hundreds of gigabytes of embeddings. This significantly saves time and reduces operations during the preprocessing stage.
>   4. (Compared to the used off-the-shelf OVD) in implementing open-vocabulary tasks, our method does not rely on the presence of a large number of unannotated novel category objects in the training dataset, reducing the requirements on the dataset.
>
>   *[1] Li, Siyuan, et al. "Matching Anything by Segmenting Anything." Proceedings of the IEEE/CVF Conference on Computer Vision and Pattern Recognition. 2024.*
> - **Q5: Will the code be public? Since this is the first work that performs end-to-end open-vocabulary object tracking, making the implementation details public is of great importance to the research community.**
>
>   Thank you very much for recognizing the value of our work. After ICLR, we will upload the revised version of the paper to arXiv and release the code.

---

### Author Response · Authors · 2024-11-27
**Common Response**

**Common Response**

First of all, we thank all Reviewers for their insightful and valuable comments. We are encouraged that the Reviewers all give the positive feedback to our work including the **first End-to-End Architecture** (Reviewer dQGj, tyWH, xFtV), **well-founded motivation** (Reviewer tyWH
, 3Y3k, xFtV), and **competitive OV-tracking performance** (Reviewer dQGj, tyWH, xFtV).

And then, we sincerely thank all Reviewers for their valuable suggestions and questions. Before formally addressing each reviewer’s comments individually, we would like to first make the following clarifications and statements.

- **OVTR is a novel paradigm of open-vocabulary MOT that achieves OV tracking in a fully end-to-end manner.** The motivation behind this work is to establish a novel tracking paradigm distinct from the mainstream OVD-based tracking architectures. This stems from the fact that, while existing OVD-based architectures demonstrate decent tracking performance, they also have several inherent limitations. These include the need for an additional OV-Detector, complex post-tracking processing, and a lack of scalability to larger datasets, which collectively hinder further improvements and real-world applications.

  In contrast, our proposed OVTR exhibits several advantageous characteristics: a single model, fully end-to-end processing, no post-processing, and a data-friendly nature that facilitates scalability. Thus, although our current approach does not significantly outperform existing OVD-based solutions at this stage, it has already demonstrated immense potential and offers the OV-MOT community a novel research direction to explore.
- **Updates to the Manuscript.** Taking into account the reviewers’ comments and suggestions for this work, we have made the following updates to the manuscript.

  1. Add more details in the Overview of the methods.
  2. Refine and correct some figures, statements and equations of methods (Sec 3.2, 3.3, 3.4, and 3.5).
  3. Update experiment parts, including more experimental setting details, SOTA performance comparisons, and ablation studies.
  4. Provide additional ablation and speed test experiments (Appendix B, beginning from line 732).
  5. Provide more details about data augumentation (Appendix C, beginning from line 805).
  6. Provide model and training hyperparameters (Appendix D, beginning from line 896).
  7. Provide more visualizations analysis (Appendix E, beginning from line 912).
  8. Provide additional preliminaries of OVTR (Sec 3.1, beginning from line 158).

  All changes in the manuscript are highlighted in red.

Finally, we have provided responses to each comment and question from the each reviewer, hoping to address your concerns adequately.

---

### Author Response · Authors · 2024-12-04
**Summary of the Discussion**

# Summary of the Discussion

We would like to express our sincere gratitude to all four reviewers for their thorough review and valuable comments.

Our method has been consistently recognized by all reviewers in the Official Review for its **novelty** or **well-founded motivation**. During the discussions with the reviewers, our responses were well-received by dQGj, tyWH, and xFtV. Consequently, **our ratings improved** to **8** (dQGj), **6** (tyWH), **6** (xFtV), and **3** (3Y3k), compared to the initial scores of **8** (dQGj), **5** (tyWH), **5** (xFtV), and **3** (3Y3k).

We have carefully addressed all the reviewers' questions with detailed explanations and revisions, which have **been acknowledged and accepted by reviewers dQGj, tyWH, and xFtV**, all of whom increased their ratings during the rebuttal stage. Especially for the issues in the method section that were partially difficult to understand, we have provided detailed explanations and corresponding revisions.

Regarding Reviewer 3Y3k, **we have thoroughly, comprehensively, and meticulously addressed all 12 (11+1) questions** raised by Reviewer 3Y3k, covering both the detailed aspects of our method and the experimental results and analyses, **as evidenced by our detailed 9-page response**. Moreover, many shared **critical and core explanations and revisions** provided in our rebuttal **have been acknowledged and accepted by the other three reviewers**. And many of these questions were related to the underlying prior knowledge of our approach, which we clarified by adding OVTR-related background in the overview and providing comprehensive responses.

Please allow us to once again summarize the advantages of our method as follows:

We propose **the first end-to-end open-vocabulary multi-object tracking (OVMOT) algorithm**, introducing **a fresh perspective** on end-to-end processing and modality interaction fusion in the OVMOT field. This approach **offers** the OVMOT community **a new research direction** to explore. We believe that **many new methods and ideas could enhance our work**, offering **substantial potential for improvement**. Furthermore, we are confident that there is **considerable room for performance enhancement**, as demonstrated by the improvements brought by our adjusted training strategy.

Specifically, we propose the CIP strategy, which enables category information to propagate across multiple frames, enhancing both tracking and classification stability. We also introduced a dual-branch decoder to improve open-vocabulary performance, **eliminating the need for generating hundreds of gigabytes of embeddings** containing category information. Additionally, the inclusion of a decoder protection strategy further boosts performance. Due to the end-to-end nature of our approach, we **eliminate the need for complex post-processing and explicit similarity associations** between frames, leading to **faster inference speeds** compared to OVTrack. Furthermore, our method does not require an additional pre-existing OV-detector. It also reduces dataset requirements and, as a transformer-based tracker with a data-friendly nature, facilitates scalability while **leaving room for further improvement**.


Finally, we would like to clarify that, in line with ICLR's policy on paper revisions, we have highlighted changes in red in the rebuttal revision. The main modifications include **adding prior knowledge** in the overview, **which may give the impression of substantial changes**. Other adjustments were made based on reviewer suggestions, and the experimental results were updated by increasing the training epochs. These revisions have been acknowledged and received by all responding reviewers, including dQGj, tyWH, and xFtV.

We genuinely hope that the reviewers and ACs will take the above points into account in the further discussion and decision-making process.

---

### Meta-Review · Area_Chair_VYGn · 2024-12-20

**Metareview:**

This paper received comments from 4 reviewers and three of them have a positive comment on this paper, but the rest (i.e., reviewer 3Y3k) have many concerns. The authors have provided a comprehensive response to the reviewer, seemingly addressing all the issues raised. The reviewer has not engaged in further in-depth communication or discussion; therefore, based on the positive feedback from all reviewers, the paper appears to meet the standards for publication.

**Additional Comments On Reviewer Discussion:**

The authors have provided detailed responses to the reviewers' questions, essentially resolving all the issues raised.

---

### Decision · Program_Chairs · 2025-01-22

Accept (Poster)